# TOWARDS FAIR KNOWLEDGE DISTILLATION USING STUDENT FEEDBACK

## ABSTRACT

With the advent of large-scale foundation models and their success in diverse fields, Knowledge Distillation (KD) techniques are increasingly used to deploy them to edge devices with limited memory and computation constraints. However, most KD works focus on improving the prediction performance of the student model distilled from large teacher models, and there is little to no work in studying the effect of distillation on key fairness properties, ensuring trustworthy distillation. In this work, we propose a fairness-driven distillation framework, BIRD (BIas-awaRe Distillation), which introduces a FAIRDISTILL operator to collect feedback from the student through a meta-learning-based approach and selectively distill teacher knowledge. We demonstrate that BIRD can be augmented with different KD methods to increase the performance of a wide range of foundation models and convolutional neural networks after distillation. Extensive experiments across three fairness datasets show the efficacy of our framework over existing state-of-the-art KD methods, opening up new directions to develop trustworthy distillation techniques.

## 1 INTRODUCTION

Recent years have witnessed an alarming trend toward developing large-scale foundation models (FMs) (Radford et al., 2021; Singh et al., 2022) trained on large datasets from unvetted data sources, leading to several deployment and fairness issues (Agarwal; Birhane et al., 2021; Mehrabi et al., 2021; Naik & Nushi, 2023; Seth et al., 2023). To address the deployment constraints of FMs, model compression techniques like Knowledge Distillation (KD) (Hinton et al., 2015) are recently used to reduce their parameter size while preserving their predictive prowess (Hsieh et al., 2023; Wang et al., 2022; Sanh et al., 2019), where KD frameworks distill knowledge from the output representations and/or logits of a teacher FM into a smaller student model. However, none of the existing works address the fairness problems of using KD in foundation models. As the distillation of large models becomes increasingly prevalent, it is essential that the resulting student models are safe and reliable, where they do not learn discriminatory features and exacerbate the bias of the teacher model.

In contrast to supervised learning frameworks where model bias is attributed to the training dataset and/or algorithm (Agarwal et al., 2021; Hooker, 2021; Yucer et al., 2022), bias in KD is influenced by the bias in the dataset, bias in the pre-trained teacher, and the nature of the KD method to optimize the student (since student models are prone to inherit teacher biases). While utilizing fair teachers may address this issue a bit, training fair teacher foundation models is an infeasible solution as i) removing biased data or using fairness objectives during teacher training conflicts with the accuracy goals of FMs, dissuading researchers from adopting them, ii) retraining existing FMs like CLIP, Flava, and GPT for fairness is hindered by limited access of original training data and extensive computational requirements, iii) the unavailability of large fairness datasets encompassing multiple bias attributes exacerbates this issue, and iv) most FMs are shielded behind APIs and intellectual property protections, limiting the transparency of their architectures and legal constraints on modifications. These contrasting aspects of the distillation framework in FMs make the problem of fair KD non-trivial.

While several existing works focus on either KD or fairness, there is little to no research on addressing them simultaneously. Despite extensive efforts in these two fields, it remains unaddressed *"what"* teacher features distill more bias during KD. To this end, Jung et al. (2021) propose MMD-based distillation, which enforces fairness constraints solely on the student model. However, they fail to provide any results to show whether it removes the bias in the teacher features or scales to large-scale

foundation models. Further, Chai et al. (2022) leverage KD to ensure fairness without using demographic labels. In contrast to existing works, we propose a joint transformation of the biased teacher knowledge and debiasing the student model with fairness objectives, where our framework learns to exclude the biased features from the teacher representations by incorporating *student feedback* via meta-learning, controlling *"what"* and *"how much"* a student distills knowledge from a teacher.

**Present Work.** We present BIRD, a novel BIas-awaRe Distillation framework that can be integrated with any existing KD framework to learn fair and accurate student representations. We first identify a key connection between the bias induced by the teacher and the bias inherited by the student (refer to Q1 in Sec. 5.2) and show that there is a trade-off between the predictive and biased knowledge distilled by a student in a KD framework. We demonstrate that existing KD frameworks result in a student model that also inherits the bias present in the teacher predictions. Further, we introduce a new fair-distillation operator that *selects* and *filters* a subset of uncorrelated features from the teacher for knowledge distillation (Sec. 4.1). The fair-distillation operator is updated using the meta-gradients from the student objective functions (Sec. 4.2). To the best of our knowledge, the proposed BIRD framework is the first to tackle the problem of fairness in KD using student feedback in a meta-learning pipeline.

We conduct extensive experimentation with three benchmark fairness datasets and several baseline KD and self-distillation techniques to analyze the efficacy of the proposed framework. Our empirical studies across different datasets and baselines reveal the following key findings. 1) The proposed fair knowledge distillation framework achieves more effective debiasing for KD compared to existing techniques. 2) BIRD is model-agnostic and can be integrated with diverse foundational models and CNNs across both knowledge-distillation and self-distillation applications. 3) BIRD introduces a simple, flexible, and computationally inexpensive fair-distillation operator trained using meta-learning that selectively distills fair and accurate teacher features. 4) Results show that BIRD improves the fairness of the knowledge distillation framework by 40.91% (on average across multiple datasets and teacher-student pairs) compared to existing MFD baselines without sacrificing predictive performance.

## 2 RELATED WORKS

This work lies at the intersection of fairness and knowledge distillation. Below we discuss related work for each of these topics.

**Knowledge Distillation.** Knowledge distillation (KD) pertains to the group of algorithms that transfers knowledge from one model to another, usually from a larger teacher model to a smaller student. While initial KD techniques predominantly focused on distilling knowledge from logits (Hinton et al., 2015), Romero et al. (2015) proposed a two-stage training procedure to perform distillation using the features of the teacher model. Following the success of Romero et al. (2015), several other variants were proposed, such as transferring attention maps (Zagoruyko & Komodakis, 2017), defining the distilled knowledge from a teacher model as the flow of the solution process (FSP) (Yim et al., 2017), and using the mutual relations between the convolution activations to perform Relational Knowledge Distillation (Park et al., 2019). In contrast to traditional KD methods that consider smaller student models than the teacher, recent works (Furlanello et al., 2018; Hahn & Choi, 2019) propose *self-distillation*, utilizing the same student-teacher architecture. In addition, recent works (Park et al., 2021; Liu et al., 2021; Zhou et al., 2021) propose new KD pipelines, where the teacher is trained using the student feedback so that *"it learns to teach,"*, thus resulting in a more holistic teacher-student framework. However, we note that all existing methods focus on improving students' predictive performance while ignoring the impact of distillation on its fairness properties, which is the motivation of this work.

**Fairness.** With the increase in the scale of large foundation models (*e.g.,* CLIP (Radford et al., 2021), FLAVA (Singh et al., 2022), etc.), it is essential to ensure that these models make fair and trustworthy decisions. Existing debiasing techniques can be categorized into methods that: i) *remove the bias* from the training dataset (Creager et al., 2019; Zemel et al., 2013; Louizos et al., 2016), ii) *apply fairness constraints* while training (Agarwal et al., 2018; Jiang & Nachum, 2020; Kamishima et al., 2012), or iii) *operate on the predicted labels* post-training (Hardt et al., 2016; Alghamdi et al., 2020). However, these approaches are designed primarily for models trained only using ground truth labels. In contrast, fairness in KD (Ahn et al., 2022) is influenced by the bias in the dataset, bias stored in the pre-trained teacher, and the nature of the KD method to optimize the student. Further, it is unclear whether we can jointly perform fair and accurate distillation or if there exist trade-offs between them. To this end, Jung et al. (2021) propose a maximum mean discrepancy (MMD) based loss function to address the fairness

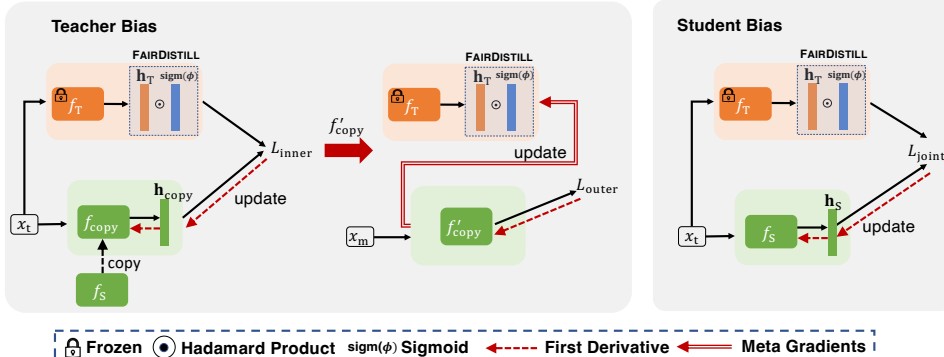

Figure 1: **Overview of BIRD framework.** BIRD learns bias-aware representations from the teacher $f_T$ by training the FAIRDISTILL operator using a meta-learning framework: **a)** for addressing teacher bias, we update FAIRDISTILL operator using $\mathcal{L}_{inner}$ and $\mathcal{L}_{outer}$ and **b)** the student model $f_S$ selectively distills unbiased representations using the formerly updated FAIRDISTILL and $\mathcal{L}_{joint}$ objective.

problem in KD. However, we note that none of the existing works adapt the incoming biased teacher knowledge based on student feedback, which can lead to a fairer student model. In contrast, our unifying framework explores the paradigm of *"learning to teach fairly"* and leads to fairer student models.

## 3 PRELIMINARIES

**Notation.** Let $\mathbf{D} = \{(\mathbf{x}_1, y_1), \ldots, (\mathbf{x}_N, y_N)\}$ be the dataset, where an image sample $\mathbf{x}_i \in \mathbb{R}^{3 \times h \times w}$ has height $h$ and width $w$, and each label $y_i \in \{1, 2, \ldots, C\}$ represents one of the $C$ classes in $\mathbf{D}$. In addition to the ground-truth label, each sample comprises a protected attribute label $y_p$ which may be used unfairly against the subject in the image. Following previous meta-learning works (Finn et al., 2017), we split the dataset $\mathbf{D}$ into three mutually exclusive sets $\mathbf{D}^{train}$, $\mathbf{D}^{test}$, and $\mathbf{D}^{meta}$.

**Knowledge Distillation.** Let $f_T$ and $f_S$ denote the teacher and student model parameterized by $\theta_T$ and $\theta_S$, respectively. We denote the output logits generated by these models as $\mathbf{z}_T$ and $\mathbf{z}_S$. The knowledge distillation loss for a given pair of teacher-student models is then defined as:

$$\mathcal{L}_{KD} = \alpha\tau^2 \sum_{i=1}^{C} \text{KL}\left(\frac{\exp(\mathbf{z}_{T,i}/\tau)}{\sum_{j=1}^{C}\exp(\mathbf{z}_{T,j}/\tau)}, \frac{\exp(\mathbf{z}_{S,i}/\tau)}{\sum_{j=1}^{C}\exp(\mathbf{z}_{S,j}/\tau)}\right) + (1-\alpha)\mathcal{L}_{CE}(\hat{\mathbf{y}}, y) \quad (1)$$

where $\hat{\mathbf{y}}$ is the softmax output of the student model $f_S$, $\text{KL}(\cdot)$ denotes the KL-Divergence loss, $\tau$ is the temperature hyperparameter in softmax, $\alpha$ is a regularization coefficient, and $\mathcal{L}_{CE}$ is the cross-entropy loss function. In addition to matching logits (Eqn. 1), *feature matching* is widely used to achieve KD. Let the output representations of the teacher and student model be $\mathbf{h}_T$ and $\mathbf{h}_S$. Feature KD (FKD) (Zagoruyko & Komodakis, 2017; Romero et al., 2015) is performed by optimizing $\mathcal{D}(\mathbf{h}_S, \mathbf{h}_T)$, where $\mathcal{D}$ is any distance metric (*e.g.,* Euclidean). In our work, we define Feature KD as:

$$\mathcal{L}_{FKD} = \alpha\mathcal{D}(\mathbf{h}_S, \mathbf{h}_T) + (1-\alpha)\mathcal{L}_{CE}(\hat{\mathbf{y}}, y) \quad (2)$$

The above definitions of logit and feature KD (Eqns. 1-2) show that student models are only optimized for their predicted performance. Since large teacher models are often biased, these frameworks distill spurious correlations in the student, further motivating the need for a bias-free distillation framework.

## 4 OUR METHOD: BIRD

Here, we describe our proposed BIRD framework that aims to generate fair and accurate student representations. BIRD demonstrates that we can obtain a fair student model by i) eliminating the biased features in the teacher representation and ii) using fairness objectives for optimizing the student during distillation. BIRD achieves this by introducing a fairness-aware distillation operator, as well as the student copy update and meta-update objective function in the distillation.

**Problem Formulation (Bias-Aware Distillation).** *Given a dataset $\mathbf{D}^{train}$ and a biased teacher model $f_T$ optimized for predictive performance on $\mathbf{D}^{train}$, we aim to learn a student model $f_S$ whose*

*representations do not reflect any undesirable discriminatory biases (i.e., they are fair) and achieve high predictive performance (i.e., they are accurate).*

## 4.1 TEACHER BIAS

It is crucial to identify features in the teacher representation $\mathbf{h}_T$ that exhibit correlation with the protected attributes $p$ as they may potentially introduce bias when distilled to the student model. To eliminate the bias in the teacher representations during distillation, we propose FAIRDISTILL, a fairness-aware distillation operator that aims to *identify* and *distill* a subset of uncorrelated (to $p$) teacher features, *i.e., selectively distill* fair teacher features. Our proposed FAIRDISTILL operator is generic in its formulation and can be augmented with any existing knowledge distillation method. It is defined to be a computationally inexpensive operator FAIRDISTILL : $\mathbb{R}^d \to \mathbb{R}^d$, which consists of a $d$-dimensional learnable weight vector $\phi$. For a given teacher representation $\mathbf{h}_T$, FAIRDISTILL is defined as:

$$\text{FAIRDISTILL}(\mathbf{h}_T) = \text{sigmoid}(\phi) \odot \mathbf{h}_T \tag{3}$$

where $\phi \in [0,1]^d$ are the learnable weight parameters, $\odot$ is the Hadamard product, and $\text{sigmoid}$ is the non-linear activation function. We apply sigmoid to the parameters $\phi$ before performing element-wise multiplication to re-weight $\mathbf{h}_T$ based on their correlation with the protected attributes.

**Student Feedback.** We leverage meta-learning frameworks consisting of two optimization steps: an *inner-loop* and *outer-loop*, where the inner-loop learns the task-specific adaptation and the outer-loop learns about the *learning process of the inner-loop* via the respective meta-parameters. In the inner optimization loop, we create a copy $f_{\text{copy}}$ of the original student model. Using $(\mathbf{x}_t, y_t) \sim \mathbf{D}^{\text{train}}$, we obtain the penultimate layer representations $\mathbf{h}_T$ and $\mathbf{h}_{\text{copy}}$ from $f_T$ and $f_{\text{copy}}$, respectively. We then leverage the FAIRDISTILL at the current step to transform $\mathbf{h}_T$ and update $f_{\text{copy}}$ using:

$$\mathcal{L}_{\text{inner}} = \alpha \mathcal{D} \left( \mathbf{h}_{\text{copy}}, \text{FAIRDISTILL}(\mathbf{h}_T) \right) + (1-\alpha)\mathcal{L}_{\text{CE}}(\hat{\mathbf{y}}_t, y_t), \tag{4}$$

where $\mathcal{D}$ can be any distance-based metric (*e.g.,* mean-squared error) and $\hat{\mathbf{y}}_t$ is the label predicted by $f_{\text{copy}}(\mathbf{x}_t)$. Since FAIRDISTILL aims to learn to *fairly distill*, we compute the resulting fairness properties of the updated student copy, *i.e.,* $f'_{\text{copy}}$, and use them as *feedback* to update FAIRDISTILL in the meta-step. Intuitively, we update FAIRDISTILL such that $f'_{\text{copy}}$ is fair, which in turn teaches FAIRDISTILL to distill fairly. We describe the same as the meta-step in BIRD.

**FAIRDISTILL update.** In the meta step, we first sample data from the meta-subset $(\mathbf{x}_m, y_m) \sim \mathbf{D}^{\text{meta}}$ and pass it through updated $f'_{\text{copy}}$. As detailed in Eqn. 4, $\theta_{\text{copy}}$ is a function of $\phi$ which implies that the gradients for $\theta'_{\text{copy}}$ is a function of the *gradients of* $\phi$. Consequently, we use $\theta'_{\text{copy}}$ to perform meta-updates on $\phi$ using a bias-aware objective function given by:

$$\mathcal{L}_{\text{outer}} = \sum_{i=1}^{C} \max \left( \sum_{j=1}^{M} \text{abs}(\mathcal{L}_{\text{CE}}(\hat{\mathbf{y}}_i|y_p = j, y_i|y_p = j) - \mathcal{L}_{\text{CE}}(\hat{\mathbf{y}}_i, y_i)) \right), \tag{5}$$

where $M$ is the number of unique values in the given protected attribute $p$, $\hat{\mathbf{y}}_i|y_p = j$ denotes the output of the network $f'_{\text{copy}}(\mathbf{x}_m)$ such that the unprotected class label is $i$ and the label of the protected attribute is $j$. Intuitively, Eqn. 5 denotes the difference between the predictive performance of $f'_{\text{copy}}$ conditioned on the unprotected/task attribute and the predictive performance of $f'_{\text{copy}}$ conditioned on both the protected attribute and given task attribute. The gradients of $\mathcal{L}_{\text{outer}}$ *w.r.t.* $\phi$ imply that $\phi$ is updated such that $f'_{\text{copy}}$ exhibits equal predictive performance across all protected demographic groups for every task category (*i.e.,* it is fair). In Fig. 1, we show the gradient flow using the meta-gradients obtained from $\mathcal{L}_{\text{outer}}$, where the gradients of $\mathcal{L}_{\text{outer}}$ *w.r.t.* $\phi$ are backpropagated via $\theta'_{copy}$ and $\phi$ is updated by computing the *meta-gradients*.

## 4.2 STUDENT BIAS

Distilling the entire teacher representation (including the biased features) and training only using predictive objectives are two leading causes of the observed bias in the student model. To learn student representations that are invariant to the protected attribute and tackle the aforementioned facets, we train BIRD using the following objectives: i) we gather feedback from $f_S$ in the form of meta-gradients to learn an optimal FAIRDISTILL (see Sec. 4.1) and ii) we apply the learned FAIRDISTILL on $f_T$ to selectively perform fair knowledge distillation. In addition, we use an explicit

model-agnostic regularization on $f_S$ that further penalizes student bias. Finally, the joint objective loss which updates the original student model $f_S$ using the *updated* FAIRDISTILL is given as:

$$\mathcal{L}_{\text{joint}} = \alpha \mathcal{D}(\mathbf{h}_S, \text{FAIRDISTILL}(\mathbf{h}_T)) + (1 - \alpha)\mathcal{L}_{\text{CE}}(\hat{\mathbf{y}}_t, y_t) + \lambda \mathcal{L}_{\text{reg}}, \tag{6}$$

where $\mathcal{L}_{\text{reg}}$ is the regularization on $f_S$ that penalizes student bias, $\lambda$ is a regularization weight for $\mathcal{L}_{\text{reg}}$, and $\hat{\mathbf{y}}_t$ is the softmax output of $f_S(\mathbf{x}_t)$. We use Eqn. 5 as the regularization term in our BIRD framework. Algorithm 1 summarizes the overall training framework of BIRD.

---

**Algorithm 1** BIRD: BIas-awaRe Distillation.

---

1: **procedure** BIRD$(\theta_S, \theta_T, \phi)$                                  ▷ Input Parameters
2:      Hyperparameters: $\mu_1, \mu_2, \mu_3$                           ▷ Learning Rates
3:      Dataset: $\mathbf{D}^{\text{train}}, \mathbf{D}^{\text{meta}}$                         ▷ Data Splits
4:      **for** $i = 1$ to num_epochs **do**                    ▷ For every epoch
5:           **while** $\mathbf{D}^{\text{train}}$ **do**                ▷ For each batch in train data
6:               $\theta_{\text{copy}} \leftarrow \theta_S$               ▷ Save current student state
7:               $\theta'_{\text{copy}} \leftarrow \theta_{\text{copy}} - \mu_1 \nabla_{\theta_{\text{copy}}} \mathcal{L}_{\text{inner}}(f_{\text{copy}})$   ▷ Update with $(\mathbf{x}_t, y_t) \sim \mathbf{D}^{\text{train}}$ using Eqn. 4
8:               $\phi \leftarrow \phi - \mu_2 \nabla_{\phi} \mathcal{L}_{\text{outer}}(f'_{\text{copy}})$    ▷ Update with $(\mathbf{x}_m, y_m) \sim \mathbf{D}^{\text{meta}}$ using Eqn. 5
9:               $\theta_S \leftarrow \theta_S - \mu_3 \nabla_{\theta_S} \mathcal{L}_{\text{joint}}(f_S)$    ▷ Update with $(\mathbf{x}_t, y_t) \sim \mathbf{D}^{\text{train}}$ using Eqn. 6
10:          **end while**

---

## 5 EXPERIMENTS

Next, we present the experimental results for our BIRD framework. We address the following key questions: Q1) Does knowledge distillation worsen/improve the bias in a student? Q2) Can we selectively distill from the teacher to ensure bias-free distillation? Q3) Can BIRD be augmented with existing knowledge distillation baselines? Q4) How do meta gradients from student models improve debiasing? Q5) Are changes to BIRD 's objective function necessary for fair predictions?

### 5.1 DATASETS AND EXPERIMENTAL SETUP

**Datasets.** We evaluate BIRD on two widely-used fairness datasets and a synthetic dataset. 1) CelebA (Liu et al., 2015) dataset comprises more than 200,000 images with 40 binary attribute annotations. Following Quadrianto et al. (2019) and Jung et al. (2021), we only consider the binary protected group and binary task class in our experiment, namely, we set Gender (male/female) as the protected attribute and Attractive (yes/no) as the target variable. 2) UTKFace (Zhang et al., 2017) dataset consists of approximately 20,000 face images with annotations of age (from 0 to 116), gender (male/female), and ethnicity (White, Black, Asian, and Indian). Images in the dataset are diverse and encompass different variations in pose, facial expression, illumination, occlusion, resolution, etc. We follow the setup described by Jung et al. (2021) and use ethnicity as the protected attribute with four classes and age as the task attribute bucketed into three classes. The synthetic dataset is the CIFAR-10S (Wang et al., 2020) dataset, which is a modified version of CIFAR-10 to study bias mitigation in image classification and consists of 32×32 images categorized into one of 10 classes. For space constraints, details and results of this synthetic dataset are in Appendix Sec. A.

**Evaluation metrics.** We report AUROC and F1-score on the test set to evaluate the predictive performance of the student. For fairness, we use two types of difference of equalized odds (DEO) metrics as proposed by Jung et al. (2021), defined upon taking the *maximum* and the *average* over the given prediction label $\hat{y}$. Further, $\Delta_{\text{max-DEO}}$ denotes the worst-case unfairness performance and $\Delta_{\text{mean-DEO}}$ represents the overall fairness across all classes. Mathematically, these metrics are defined as:

$$\Delta_{\text{mean-DEO}} = \frac{1}{|C|} \sum_{y_i} |P(\hat{y} = y_i | y = y_i, y_p = 0) - P(\hat{y} = y_i | y = y_i, y_p = 1),| \tag{7}$$

$$\Delta_{\text{max-DEO}} = \max_{y_i} |P(\hat{y} = y_i | y = y_i, y_p = 0) - P(\hat{y} = y_i | y = y_i, y_p = 1),| \tag{8}$$

where $\hat{y}$ represents the predicted label, $y_p$ is a given protected attribute, and $y_i$ denotes a label from all possible labels $C$ in the dataset.

**Baseline methods.** We consider the standard knowledge distillation baselines: **Base KD (BKD)** (Hinton et al., 2015), and **FitNet** (Romero et al., 2015) – they entirely focus on improving student's prediction accuracy. In addition, we include recent methods proposed to tackle fairness in KD: **Adversarial debiasing** (Zhang et al., 2018), **MFD** (Jung et al., 2021). Further, in our experiments with CNNs in the Appendix, we include another kd baseline - **Attention Transfer** (Zagoruyko & Komodakis, 2017) that requires access to the spatial features of the teacher network.

**Model architectures.** We investigate the flexibility of BIRD using three established foundation models (FMs): CLIP-ResNet-50, CLIP-ViT-B/32 (Radford et al., 2021), and FLAVA (Singh et al., 2022). In addition, we consider three widely used Convolutional Neural Network (CNN) architectures in knowledge distillation to show the generalizability of BIRD in performing bias-aware distillation: ShuffleNetV2 (Ma et al., 2018), ResNet-18, and ResNet-34 (He et al., 2016). We use the public implementations and pre-trained weights for FMs and CNNs models from HuggingFace (Wolf et al., 2020) and PyTorch model-zoo (PyTorch), respectively. Note that while we initialize the FMs using their pre-trained weights, the CNN models were trained from scratch.

**Baseline Implementation.** We use Adam optimizer (Kingma & Ba, 2015) with its default parameters and a learning rate of $1 \times 10^{-3}$ to train all our baseline models. For the CelebA dataset, all models are trained for 10 epochs with a constant learning rate. However, for the UTK dataset, we train the CNNs for 50 epochs with a decay factor of $1 \times 10^{-1}$ every 10 epochs and FMs for 10 epochs with a constant learning rate. We follow previous works and use $\alpha = 0.90$ and $\tau = 4$ for the knowledge distillation parameters for all experiments. We implement FitNet (Romero et al., 2015) following their official paper Romero et al. (2015), AT loss using the official repository [1] with $\beta = 1 \times 10^6$, and Adversarial Debiasing from the repository[2] provided by Wang et al. (2020). Refer to the Appendix for details on baseline hyperparameters.

**BIRD Implementation.** We use PyTorch (Paszke et al., 2017) and Higher (Grefenstette et al., 2019) to implement the BIRD framework. Following previous meta-learning works (Liu et al., 2022), BIRD utilizes two mutually exclusive data splits namely, $\mathbf{D}^{\text{train}}$ and $\mathbf{D}^{\text{meta}}$ for training $f_S$ and FAIRDISTILL, respectively (also see Fig. 1). The meta subset is created by randomly sampling $20\%$ examples from the training dataset (similar to (Finn et al., 2017)). The baselines are trained using the entire set of training examples $\mathbf{D}^{\text{train}} \cup \mathbf{D}^{\text{meta}}$. We use SGD optimizer (Ruder, 2016) with a momentum of $0.90$ and weight decay of $5 \times 10^{-4}$ to train $f_S$ because it is non-trivial to achieve convergence using Adam optimizer in performing meta-optimization. The $\phi$ parameters are updated using AdamW optimizer with a weight decay of $5 \times 10^{-2}$, learning rate of $1 \times 10^{-3}$ after a fixed number of warmup epochs to ensure that $f_S$ is partially optimized for KD and can generate meaningful meta-gradients. Finally, with careful analysis and linear probing, we find the optimal $\lambda$ weight hyperparameter to regularize fairness (see Eqn. 6) for each architecture setting and show the individual values in the Appendix. We use a single A100 GPU with 80GB GPU memory for our experiments.

## 5.2 RESULTS

Here, we discuss our experimental results that address the aforementioned questions Q1-Q5.

**1) KD introduces bias in student models.** To demonstrate the impact of an unfair teacher, we compute the DEO metrics (both mean and max) for a student model before and after distillation. We use BKD (refer to Eqn. 1) with CLIP-ViT-B/32 and FLAVA models as teachers and the ResNet-{18,34} as students. Across different teacher-student combinations, results in Fig. 2 show that the student model, which originally had better fairness performance, becomes unfair (higher metric values) after inheriting the teacher's biased features. On average, we find an increase of 38.54% in $\Delta_{\text{mean-DEO}}$ and 37.26% in $\Delta_{\text{max-DEO}}$, reinforcing our hypothesis that vanilla KD introduces bias in the student model. See the Appendix for similar insights on additional teacher-student architectures.

**2) BIRD improves the fairness of knowledge distillation.** A vital property of any KD framework is to effectively and fairly distill from a given unfair teacher model. We conduct multiple experiments on different teacher-student combinations to analyze the efficacy of BIRD over different KD methods. Specifically, we present our results for both Self Distillation (similar to Jung et al. (2021)) and Knowledge Distillation settings in Table 1 and Table 2 respectively. Across multiple datasets and

---

[1]`https://github.com/szagoruyko/attention-transfer`
[2]`https://github.com/princetonvisualai/DomainBiasMitigation`

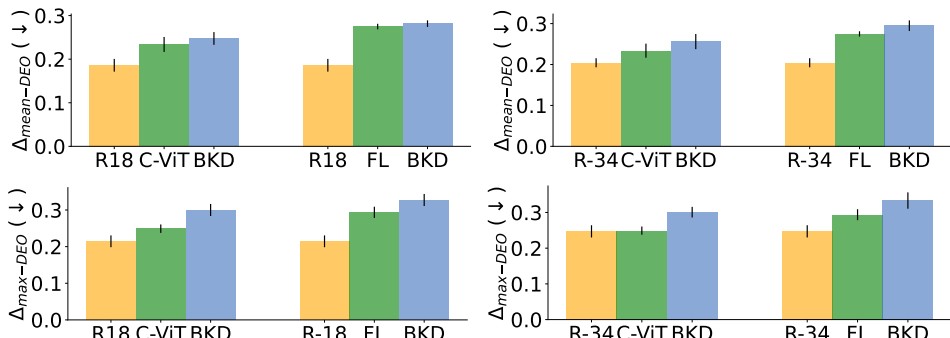

Figure 2: Difference of Equalized Odd (DEO) metric scores for baseline teacher (CLIP-ViT-B/32 (C-Vit), FLAVA (FL)), baseline student (ResNet-{18, 34}), and distilled student models using base knowledge distillation (BKD). We find that knowledge distillation results in unfairer student predictions as compared to baseline students across both $\Delta_{\text{mean-DEO}}$ (row 1) and $\Delta_{\text{max-DEO}}$ (row 2) metrics.

Table 1: Results of self-distillation on foundation models using two fairness datasets. Shown is the average performance across five independent runs. Arrows ($\uparrow$, $\downarrow$) indicate the direction of better performance. BIRD retains the predictive power (AUROC and F1-score) of the baseline model while improving their fairness (shaded area). Note that teacher-student architectures in each case are identical. See Appendix for fairness results of widely used CNNs in KD and CIFAR-10S results.

| Model | Dataset | Method | AUROC ($\uparrow$) | F1-score ($\uparrow$) | $\Delta_{\text{mean-DEO}}(\downarrow)$ | $\Delta_{\text{max-DEO}}(\downarrow)$ |
|---|---|---|---|---|---|---|
| CLIP-ViT-32 | UTKFace | Baseline | 95.96±0.03 | 86.22±0.16 | 13.47±0.20 | 25.07±0.96 |
| | | BKD | 95.95±0.05 | 86.02±0.15 | 13.73±0.15 | 25.27±0.99 |
| | | FitNet | 95.95±0.06 | 86.07±0.29 | 13.80±0.21 | 25.47±1.42 |
| | | AD | 96.05±0.06 | 86.34±0.32 | **11.84**±0.73 | 22.49±0.31 |
| | | MFD | 96.05±0.04 | 86.64±0.15 | 12.11±0.16 | 22.79±0.37 |
| | | **BIRD** | 95.50±0.04 | 85.67±0.08 | 12.07±0.27 | **16.92**±0.82 |
| | CelebA | Baseline | 87.01±0.26 | 78.15±0.52 | 23.38±1.72 | 24.91±1.15 |
| | | BKD | 87.07±0.26 | 78.20±0.48 | 23.26±1.67 | 24.62±1.14 |
| | | FitNet | 87.08±0.23 | 78.16±0.46 | 23.25±1.50 | 24.57±1.02 |
| | | AD | 88.20±0.17 | 79.00±0.15 | 17.02±1.03 | 17.82±0.97 |
| | | MFD | 87.22±0.11 | 77.59±0.70 | 21.99±0.70 | 23.70±1.58 |
| | | **BIRD** | 88.55±0.03 | 80.84±0.06 | **3.44**±0.92 | **5.19**±1.06 |
| CLIP-ResNet50 | UTKFace | Baseline | 95.67±0.04 | 84.90±0.20 | 13.70±0.58 | 23.08±1.06 |
| | | BKD | 95.67±0.03 | 84.85±0.20 | 13.57±0.50 | 23.28±0.99 |
| | | FitNet | 95.66±0.03 | 84.98±0.28 | 13.93±0.45 | 23.78±1.01 |
| | | AD | 95.67±0.05 | 83.86±0.16 | 14.83±1.58 | 26.27±0.70 |
| | | MFD | 95.69±0.03 | 84.90±0.52 | 14.16±0.60 | **22.99**±1.60 |
| | | **BIRD** | 95.43±0.02 | 84.05±0.13 | **12.43**±0.14 | 23.28±0.43 |
| | CelebA | Baseline | 87.72±0.06 | 78.71±0.21 | 21.11±0.30 | 21.97±0.41 |
| | | BKD | 87.72±0.06 | 78.90±0.15 | 21.10±0.40 | 22.07±0.41 |
| | | FitNet | 87.75±0.06 | 78.77±0.21 | 21.00±0.28 | 21.80±0.38 |
| | | AD | 88.51±0.02 | 80.32±0.05 | 5.33±0.19 | 7.93±0.22 |
| | | MFD | 87.49±0.12 | 78.56±0.21 | 22.56±0.56 | 23.52±0.33 |
| | | **BIRD** | 87.93±0.01 | 80.34±0.08 | **2.65**±0.29 | **4.49**±0.48 |

state-of-the-art foundational models, we show that BIRD learns fairer student representations while preserving the predictive performance of the original model over strong baselines. On average across both datasets and models, BIRD improves the fairness of the underlying model by 46.29% (in $\Delta_{\text{mean-DEO}}$) and 44.33% (in $\Delta_{\text{max-DEO}}$), respectively as compared to MFD for the self distillation setting. Similarly, BIRD improves the fairness of the underlying model by 38.47% (in $\Delta_{\text{mean-DEO}}$) and 34.54% (in $\Delta_{\text{max-DEO}}$), respectively for the knowlegde distillation setting. We observe that BIRD consistently achieves the best fairness performance across all methods without significantly impacting their predictive performance for both self and knowledge distillation framework. We also analyze the efficacy of BIRD across widely used CNNs in distillation literature (see Table 4).

Table 2: Results of knowledge distillation using foundation models, ResNets, and two fairness datasets. Shown is the average performance across five independent runs. Arrows ($\uparrow$, $\downarrow$) indicate the direction of better performance. BIRD retains the predictive power (AUROC and F1-score) of the baseline model while improving their fairness (shaded area).

| Model | Dataset | Method | AUROC ($\uparrow$) | F1-score ($\uparrow$) | $\Delta_{\text{mean-DEO}}(\downarrow)$ | $\Delta_{\text{max-DEO}}(\downarrow)$ |
|---|---|---|---|---|---|---|
| CLIP-ViT-32 $\rightarrow$ResNet18 | CelebA | Student | $85.44\pm0.29$ | $74.26\pm1.59$ | $18.60\pm1.46$ | $21.46\pm1.61$ |
| | | Teacher | $87.01\pm0.26$ | $78.15\pm0.52$ | $23.38\pm0.20$ | $24.91\pm1.15$ |
| | | BKD | $87.03\pm0.28$ | $77.55\pm0.94$ | $25.04\pm1.46$ | $28.81\pm1.53$ |
| | | FitNet | $87.52\pm0.32$ | $78.85\pm0.52$ | $23.96\pm1.75$ | $26.29\pm1.38$ |
| | | AD | $79.50\pm1.95$ | $71.61\pm1.81$ | $8.48\pm3.55$ | $11.30\pm3.21$ |
| | | MFD | $87.06\pm0.05$ | $77.86\pm0.28$ | $18.31\pm1.60$ | $25.27\pm0.84$ |
| | | **BIRD** | $89.26\pm0.06$ | $80.86\pm0.15$ | $\mathbf{7.51}\pm1.03$ | $\mathbf{10.16}\pm1.12$ |
| | UTKFace | Student | $92.25\pm0.14$ | $78.73\pm0.27$ | $17.21\pm0.40$ | $36.92\pm1.13$ |
| | | BKD | $95.06\pm0.05$ | $81.12\pm0.56$ | $17.74\pm0.82$ | $36.22\pm0.82$ |
| | | FitNet | $94.76\pm0.07$ | $81.59\pm0.52$ | $16.52\pm0.69$ | $33.63\pm1.81$ |
| | | AD | $86.55\pm1.82$ | $69.03\pm0.57$ | $30.55\pm2.19$ | $50.85\pm4.55$ |
| | | MFD | $90.45\pm0.35$ | $77.61\pm0.25$ | $18.31\pm1.60$ | $34.73\pm2.21$ |
| | | **BIRD** | $91.05\pm0.15$ | $77.71\pm0.61$ | $\mathbf{14.96}\pm1.23$ | $\mathbf{30.55}\pm2.68$ |
| CLIP-R50 $\rightarrow$ResNet18 | CelebA | Student | $85.44\pm0.29$ | $74.26\pm1.59$ | $18.60\pm1.46$ | $21.46\pm1.61$ |
| | | BKD | $87.63\pm0.15$ | $78.72\pm0.47$ | $22.99\pm0.47$ | $26.25\pm1.39$ |
| | | FitNet | $87.61\pm0.1$ | $81.59\pm0.52$ | $22.93\pm0.32$ | $26.51\pm0.8$ |
| | | AD | $81.41\pm1.59$ | $62.85\pm3.21$ | $17.80\pm6.43$ | $25.52\pm9.54$ |
| | | MFD | $87.09\pm0.25$ | $77.55\pm0.37$ | $21.88\pm0.38$ | $23.52\pm0.65$ |
| | | **BIRD** | $85.72\pm0.96$ | $74.15\pm2.54$ | $\mathbf{6.20}\pm1.73$ | $\mathbf{9.83}\pm2.77$ |
| | UTKFace | Student | $92.25\pm0.14$ | $78.73\pm0.27$ | $17.21\pm0.40$ | $36.92\pm1.13$ |
| | | BKD | $95.25\pm0.08$ | $82.51\pm0.62$ | $16.85\pm0.86$ | $35.52\pm1.07$ |
| | | FitNet | $94.93\pm0.16$ | $81.42\pm0.38$ | $\mathbf{16.25}\pm1.08$ | $34.33\pm2.11$ |
| | | AD | $85.50\pm1.15$ | $68.28\pm1.88$ | $28.46\pm1.46$ | $53.33\pm2.72$ |
| | | MFD | $89.88\pm0.27$ | $75.87\pm1.10$ | $18.77\pm1.07$ | $39.90\pm1.99$ |
| | | **BIRD** | $90.59\pm0.13$ | $77.04\pm0.87$ | $17.84\pm1.30$ | $\mathbf{33.35}\pm1.21$ |

Table 3: Results of BIRD for three different KD methods. Shown is the average performance across five independent runs on the Celeb-A dataset with ResNet18$\rightarrow$ResNet18. BIRD consistently improves the fairness performance (shaded area) of all existing KD methods.

| Method | AUROC ($\uparrow$) | F1-score ($\uparrow$) | $\Delta_{\text{mean-DEO}}(\downarrow)$ | $\Delta_{\text{max-DEO}}(\downarrow)$ |
|---|---|---|---|---|
| FitNet | $85.99\pm0.18$ | $75.59\pm0.99$ | $19.01\pm1.17$ | $21.46\pm1.37$ |
| BIRD + FitNet (Stage 2) | $84.49\pm0.18$ | $77.04\pm0.35$ | $\mathbf{11.32}\pm1.29$ | $\mathbf{16.15}\pm1.38$ |
| AT | $86.03\pm0.20$ | $75.07\pm1.50$ | $18.00\pm1.19$ | $22.16\pm1.54$ |
| BIRD + AT | $86.92\pm0.16$ | $78.56\pm0.33$ | $\mathbf{3.55}\pm0.37$ | $\mathbf{5.24}\pm0.56$ |
| MFD | $86.24\pm0.09$ | $77.32\pm0.26$ | $19.34\pm0.47$ | $21.46\pm0.64$ |
| BIRD + MFD | $82.61\pm0.34$ | $75.20\pm0.63$ | $\mathbf{15.25}\pm0.45$ | $\mathbf{18.09}\pm0.45$ |

**3) BIRD-augmented methods are fairer than their vanilla counterpart.** We conduct multiple experiments on different BIRD-augmented knowledge distillation techniques to analyze the efficacy of BIRD. In particular, we augment BIRD with two widely used KD methods (FitNet and AT) and MFD, a baseline to achieve fairness in KD. Our results in Table 3 demonstrate that BIRD-augmented knowledge distillation techniques learn fairer representations than their vanilla counterparts. On average, BIRD improves the fairness of three existing KD methods by 41.86% (in $\Delta_{\text{mean-DEO}}$) and 41.80% (in $\Delta_{\text{max-DEO}}$), respectively. A key takeaway from our experiments is that BIRD learns a small distillation operator using meta-learning that can be easily integrated with any existing KD frameworks.

**4) Meta-updates improve fairness.** We conduct an ablation on the importance of the meta-step in our BIRD framework. The meta step allows us to update the parameters of FAIRDISTILL such that the biased features from the teacher do not impact the student. We substantiate its effectiveness by removing this operator and performing KD with only the regularization term $L_{\text{reg}}$. Results show that the meta-learning component is necessary to learn fair representations (Figure 3). In particular, we observe a 6.45% improvement in the fairness of BIRD, as compared to BIRD w/o Meta, providing empirical evidence that the meta-gradients improve the fairness of the student model.

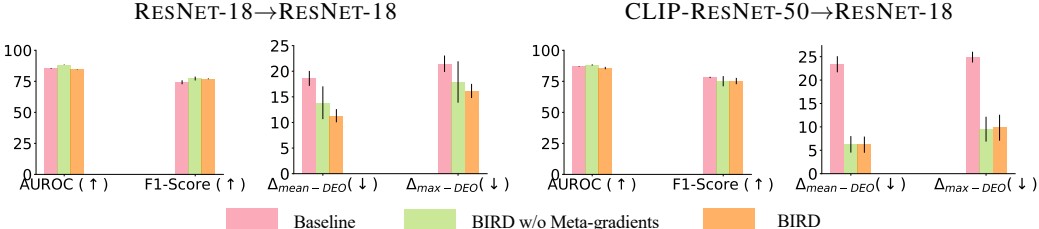

Figure 3: Ablation study to understand the impact of meta-gradients in BIRD. Shown is the average performance across five independent runs on the Celeb-A dataset with ResNet18 and CLIP-RN50 models, evidencing that the student feedback through the meta-step update improves fairness.

**5) Ablation study.** We conduct ablations on several components of the BIRD framework, namely the impact of the $\mathcal{L}_{\text{outer}}$ loss objective, computational overhead compared to baseline, and comparing BIRD with methods that use KD to ensure fairness without using demographic labels. Our ablation on the $\mathcal{L}_{\text{outer}}$ objective shows that our proposed framework is agnostic to the choice of the loss function. Results in Figure 4 show that all three variants of BIRD with different objectives achieve better predictive performance (AUROC and F1-Score) and fairness ($\Delta_{\text{mean-DEO}}$ and $\Delta_{\text{max-DEO}}$) compared to the baseline model and its vanilla KD counterpart. While BIRD-MSE achieves the best performance in our ablation using the CelebA dataset, we used the max loss objective (Eq. 5) for our main experiments as, on average, it achieves better predictive and fairness performance across all datasets and foundation models. Further, for our ablation on the computational overhead, we analyze the GPU memory reserved and the number of parameters for a ResNet18 model on both BIRD and MFD. We observe that BIRD is highly efficient in terms of GPU memory consumption and reserves 11878MiB GPU memory in comparison to MFD, which reserves 62400MiB (this overhead is due to the expensive kernel operations in the MMD loss proposed in MFD (Jung et al., 2021)). In addition, BIRD only introduces 512 (for ResNet18) new trainable parameters compared to existing methods (a 0.46% increase in parameters compared to MFD and BKD). Finally, we conduct experiments to compare the fairness performance of BIRD with a method (denoted as FWD Chai et al. (2022)) that uses KD to achieve fairness without utilizing demographic labels(see Table 6-7) for more details).

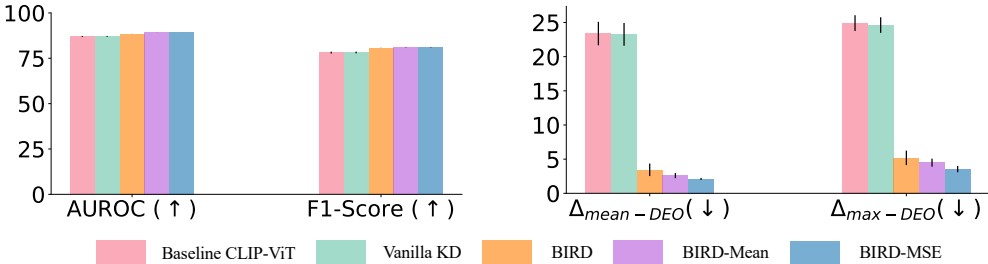

Figure 4: Results of BIRD on different $\mathcal{L}_{\text{outer}}$ losses on CelebA dataset. Shown is the average performance across five independent runs. BIRD achieves better predictive (AUROC and F1-score) and fairness ($\Delta_{\text{mean-DEO}}$ and $\Delta_{\text{max-DEO}}$) than baseline and vanilla KD models across different $\mathcal{L}_{\text{outer}}$ objectives.

## 6  CONCLUSION

The recent surge in the development of large-scale vision-language models and commercial APIs suffers from deployment and fairness issues due to their enormous parameter size and unvetted training datasets. In this work, we address the problem of learning fair distilled students. To this end, we introduce BIRD, a meta-learning framework that exploits a critical connection between *"what"* and *"how much"* knowledge to distill from a given teacher. We demonstrate that BIRD leverages important student feedback to identify and transfer teacher features uncorrelated to a given protective attribute, resulting in fairer and more accurate student representation. Our results on three benchmark datasets show that BIRD consistently improves the fairness (in terms of difference of equalized odds metric) compared to state-of-the-art knowledge distillation and debiasing techniques. This work paves the way for an exciting direction to develop trustworthy distillation techniques, where student feedback can guide the distillation process to distill trustworthy features from the teacher.

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

## A   APPENDIX

Here, we present additional results and ablations of BIRD. Particularly, we replicate the setup by Jung et al. (2021) and extend our self-distillation experiments on additional architectures (CNNs: ShuffleNet-v2, ResNet-18, ResNet-34, and FM: Flava). Further, we perform experiments on ResNet-18 using a synthetic dataset, namely, CIFAR-10S. Albeit different from the problem we are solving, we compare BIRD against an additional baseline technique Chai et al. (2022) that tackles the problem of fairness when sensitive labels are not present. In Table 8, we share the exact metrics for the illustration shown in Fig. 2 which further establishes the argument that knowledge distillation incurs bias.

**Results on additional models and CIFAR-10S.** We observe that BIRD consistently improves the fairness performance for all architectures in the CNN experiments for two real-world, widely used visual fairness datasets while maintaining their predictive performance (See Table 4). Interestingly, while AD does improve fairness in some cases (Table 4 CelebA) it fails to do so consistently while maintaining the F1 and AUROC scores. Finally, we see a significant improvement in both $\Delta_{\text{max-DEO}}$ and $\Delta_{\text{mean-DEO}}$ for Flava on both UTKFace and CelebA datasets as shown in Table 4.

We reproduce the experimental setup by Wang et al. (2020) for CIFAR-10S, and show the results for the same in Table 5. We show that BIRD is able to improve both $\Delta_{\text{max-DEO}}$ by 20.19% (47.94→38.26) and $\Delta_{\text{mean-DEO}}$ by 24.71% (26.26→19.77) over the baseline student. It is noteworthy that even AD significantly improves the fairness metrics, however, this results in a substantial drop in F1-Scores. On the other hand, BIRD obtains the best predictive performance metrics amongst all baselines.

**Additional Baseline: Fairness without Demographics.** In this section, we compare with Chai et al. (2022), a technique that uses knowledge distillation to improve fairness in a setting where demographic labels are not available. Although the problem statement is different and their approach addresses fairness without demographics, this section includes their application to the problem described in our work. For results in Table 6 and Table 7, we refer to Chai et al. (2022) as **FWD**, and we clearly observe that BIRD consistently outperforms FWD on various model configurations and datasets.

### A.1   ADDITIONAL HYPERPARAMETER DETAILS

In this section, we discuss the critical hyperparameter choices for BIRD and predominant baselines. For the CIFAR-10S dataset, we observe that the widely accepted temperature $\tau = 4$ and $\alpha = 0.90$ do not give optimal BKD performance. Thereby, we tune the strength of the aforementioned hyperparameters and use $\tau = 10$ and $\alpha = 0.50$ instead. We observe that setting the training ratio parameter (base model updates compared to domain classifier) to 3 if the total number of epochs is greater than 20, and 1 otherwise helps retain the predictive performance the best when using AD. We observe that this largely retains the predictive performance of AD. The feature distillation strength for FitNet loss in Stage 2 is kept constant at $0.1$. For the experiments conducted on foundation models (FLAVA, CLIP-ViT-32, CLIP-R50) in our paper, we operate under the assumption that only the penultimate representation layer is accessible, following a black-box setting. As a result, employing AT is not feasible, and that is why the corresponding results are excluded from Table 1. Lastly, since Jung et al. (2021) does not provide a public codebase, we try to implement MFD as faithfully as possible keeping testing conditions consistent across all our experiments. Please refer to Table 9 for the list of hyperparameters for different BIRD experiments. For experiments in Table 3, we combine BIRD with existing KD and Fair KD pipelines. For BIRD +FitNet, we apply Fitnet stage 2 loss in addition to the BIRD optimization 6 with feature distillation strength kept at $0.1$. Similarly, for BIRD +AT and BIRD +MFD, we augment Eq 6 with the attention transfer loss and MMD loss respectively keeping the rest of the bird framework intact.

Table 4: Results of self-distillation on three CNN models and an additional FM using two fairness datasets. Shown is the average performance across five independent runs. Arrows (↑, ↓) indicate the direction of better performance. BIRD retains the predictive power (AUROC and F1-score) of the baseline model while improving their fairness (shaded area).

| Model | Dataset | Method | AUROC (↑) | F1-score (↑) | $\Delta_{\text{mean-DEO}}(\downarrow)$ | $\Delta_{\text{max-DEO}}(\downarrow)$ |
|---|---|---|---|---|---|---|
| ShuffleNetV2 | UTKFace | Baseline | 89.15±0.37 | 74.05±0.56 | 20.33±1.17 | 42.29±2.42 |
| | | BKD | 90.43±0.13 | 74.60±0.23 | 20.00±0.68 | 38.31±2.13 |
| | | FitNet | 89.84±0.44 | 73.63±0.47 | 20.20±0.87 | 42.59±2.39 |
| | | AT | 90.70 ±0.29 | 76.12 ±0.44 | 19.34 ±1.74 | 40.10 ±2.58 |
| | | AD | 90.16 ±0.15 | 75.60 ±0.29 | 19.60 ±0.63 | 40.30 ±2.19 |
| | | MFD | 90.11±0.27 | 74.75±0.85 | 19.77±0.67 | 37.91 ±0.81 |
| | | **BIRD** | 90.53±0.37 | 74.88±0.61 | **16.92**±1.15 | **36.12**±1.39 |
| | CelebA | Baseline | 86.01±0.04 | 76.44 ±1.21 | 23.11±0.20 | 28.38±0.60 |
| | | BKD | 86.20±0.11 | 76.81 ±0.86 | 23.26±0.59 | 26.72±1.78 |
| | | FitNet | 85.84±0.20 | 76.93±0.40 | 22.98±0.89 | 25.17±1.76 |
| | | AT | 86.27 ±0.11 | 75.51 ±1.50 | 25.17 ±1.76 | 27.54 ±1.97 |
| | | AD | 86.51 ±0.18 | 77.64 ±0.79 | 8.04 ±1.96 | 11.18 ±2.33 |
| | | MFD | 85.88±0.08 | 76.72±0.52 | 21.59±0.39 | 23.81±1.06 |
| | | **BIRD** | 88.01±0.27 | 79.82±0.29 | **5.01**±1.04 | **8.16**±2.17 |
| ResNet18 | UTKFace | Baseline | 92.25±0.14 | 90.80±0.16 | 17.21±0.40 | 36.92±1.13 |
| | | BKD | 93.06±0.17 | 80.22 ±0.49 | 18.54±0.81 | 39.00±2.13 |
| | | FitNet | 93.01±0.17 | 79.35±0.20 | 17.42±1.10 | 34.93±2.43 |
| | | AT | 92.92 ±0.12 | 80.30 ±0.24 | 17.88 ±0.71 | 36.22 ±1.52 |
| | | AD | 90.93 ±0.46 | 78.61 ±0.47 | 17.18 ±0.73 | 36.32 ±2.32 |
| | | MFD | 93.03±0.11 | 80.10±0.19 | 16.62±0.70 | 36.22±0917 |
| | | **BIRD** | 91.67±0.29 | 77.71±0.42 | **15.49**±0.77 | **30.65**±3.42 |
| | CelebA | Baseline | 85.44±0.29 | 74.26±1.59 | 18.60±1.46 | 21.46±1.61 |
| | | BKD | 86.17±0.18 | 74.52±1.53 | 17.98±1.81 | 21.12±1.99 |
| | | FitNet | 85.99±0.18 | 75.59±0.99 | 19.01±1.17 | 21.46±1.37 |
| | | AT | 86.03 ±0.20 | 75.07 ±1.50 | 18.00 ±1.19 | 22.16 ±1.54 |
| | | AD | 60.19 ±2.88 | 55.63 ±1.69 | 13.78 ±4.54 | 16.89 ±4.42 |
| | | MFD | 86.24±0.09 | 77.32 ±0.26 | 19.34±0.47 | 21.46±0.64 |
| | | **BIRD** | 84.49±0.18 | 77.04±0.35 | **11.32**±1.29 | **5.31**±1.38 |
| ResNet34 | UTKFace | Baseline | 92.18±0.35 | 78.96±0.33 | 17.61 ±0.90 | 36.52 ±1.16 |
| | | BKD | 92.36±0.43 | 79.95 ±0.30 | 17.41±0.63 | 37.91 ±1.45 |
| | | FitNet | 92.23±0.45 | 80.15±0.29 | 17.81±0.54 | 36.12±1.99 |
| | | AT | 92.06 ±0.27 | 79.50 ±0.27 | 16.72 ±1.05 | 34.03 ±2.26 |
| | | AD | 92.08 ±0.40 | 79.05 ±0.70 | 18.04 ±1.45 | 36.52 ±3.33 |
| | | MFD | 92.48±0.16 | 78.76±0.31 | 16.98 ±0.78 | 35.42 ±1.08 |
| | | **BIRD** | 90.90±0.16 | 77.74±0.43 | **15.92**±0.55 | **33.13**±1.31 |
| | CelebA | Baseline | 85.93±0.31 | 75.73±0.53 | 20.43±1.11 | 24.71 ±1.71 |
| | | BKD | 86.32±0.36 | 77.04±0.26 | 20.88±1.43 | 23.29 ±1.83 |
| | | FitNet | 86.16±0.29 | 73.73±2.75 | 22.45±0.69 | 28.91±2.54 |
| | | AT | 85.95 ±0.38 | 74.89 ±0.80 | 21.05 ±0.79 | 25.35 ±1.59 |
| | | AD | 69.32 ±4.14 | 61.83 ±3.10 | 28.52 ±7.25 | 34.12 ±8.88 |
| | | MFD | 87.10±0.10 | 78.04 ±0.06 | 17.48 ±0.80 | 17.91 ±0.90 |
| | | **BIRD** | 84.31±0.37 | 73.90±0.55 | **10.31**±2.88 | **13.59**±3.92 |
| FLAVA | UTKFace | Baseline | 94.43±0.04 | 81.12±0.28 | 14.49±0.49 | 32.54±1.62 |
| | | BKD | 94.43±0.04 | 81.02±0.23 | 15.02±0.51 | 32.84±1.51 |
| | | FitNet | 94.33±0.04 | 81.54±0.48 | 14.93±0.37 | 32.34±1.72 |
| | | AD | 92.81 ±0.09 | 77.21 ±0.44 | 17.08 ±0.12 | 37.11 ±0.21 |
| | | MFD | 94.42±0.10 | 80.77±0.32 | 15.42±1.38 | 33.03 ±3.73 |
| | | **BIRD** | 94.00±0.02 | 80.74±0.54 | **14.23**±0.55 | **28.16**±2.89 |
| | CelebA | Baseline | 84.43±0.12 | 74.87 ±0.21 | 27.48±0.64 | 29.37±1.53 |
| | | BKD | 84.42±0.11 | 74.96±0.22 | 27.39±0.58 | 29.36±1.41 |
| | | FitNet | 84.40±0.11 | 74.94±0.60 | 27.36±0.65 | 29.23±1.50 |
| | | AD | 84.35 ±0.05 | 77.88 ±0.20 | 10.54 ±0.80 | 12.93 ±0.79 |
| | | MFD | 84.45±0.11 | 75.55±0.14 | 26.64±0.62 | 28.63±0.68 |
| | | **BIRD** | 85.48±0.02 | 78.54±0.10 | **2.53**±0.17 | **4.12**±0.59 |

Table 5: Results on CIFAR-10S dataset across 5 independent runs for ResNet18→ResNet18. Arrows (↑, ↓) indicate the direction of better performance. BIRD retains the predictive power (AUROC and F1-Score) of the baseline model while improving the fairness criterion ($\Delta_{\text{mean-DEO}}$ and $\Delta_{\text{max-DEO}}$)

| Method | AUROC (↑) | F1-score (↑) | $\Delta_{\text{mean-DEO}}$(↓) | $\Delta_{\text{max-DEO}}$(↓) |
|---|---|---|---|---|
| Baseline | $98.91_{\pm 0.02}$ | $88.34_{\pm 0.17}$ | $26.26_{\pm 0.70}$ | $47.94_{\pm 1.94}$ |
| BKD | $98.95_{\pm 0.02}$ | $88.90_{\pm 0.13}$ | $25.30_{\pm 0.63}$ | $46.92_{\pm 2.16}$ |
| FitNet | $98.89_{\pm 0.01}$ | $88.15_{\pm 0.08}$ | $26.55_{\pm 0.66}$ | $48.86_{\pm 1.85}$ |
| AT | $98.99_{\pm 0.02}$ | $88.95_{\pm 0.12}$ | $25.16_{\pm 0.33}$ | $46.08_{\pm 2.27}$ |
| AD | $98.44_{\pm 0.11}$ | $85.98_{\pm 0.43}$ | $\mathbf{16.20}_{\pm 1.18}$ | $\mathbf{31.94}_{\pm 3.89}$ |
| MFD | $98.93_{\pm 0.03}$ | $88.32_{\pm 0.10}$ | $27.27_{\pm 0.34}$ | $49.16_{\pm 1.62}$ |
| BIRD | $\mathbf{99.12}_{\pm 0.02}$ | $\mathbf{89.45}_{\pm 0.14}$ | $19.77_{\pm 0.37}$ | $38.26_{\pm 1.73}$ |

Table 6: Results of BIRD compared to FWD (Chai et al., 2022) on CelebA dataset for several architectures. BIRD consistently outperforms the additional baseline on both predictive performance and fairness metrics.

| Method | AUROC (↑) | F1 (↑) | $\Delta_{\text{mean-DEO}}$ | $\Delta_{\text{max-DEO}}$ |
|---|---|---|---|---|
| FWD (CLIP-ViT-32→Res-18) | $86.68_{\pm 0.34}$ | $77.55_{\pm 0.75}$ | $24.89_{\pm 1.48}$ | $28.32_{\pm 1.41}$ |
| BIRD (CLIP-ViT-32→Res-18) | $\mathbf{89.26}_{\pm 0.06}$ | $\mathbf{80.86}_{\pm 0.15}$ | $\mathbf{7.51}_{\pm 1.03}$ | $\mathbf{10.16}_{\pm 1.12}$ |
| FWD (CLIP-RN50→Res-18) | $87.24_{\pm 0.33}$ | $76.2_{\pm 2.1}$ | $23.27_{\pm 1.5}$ | $28.04_{\pm 1.67}$ |
| BIRD (CLIP-RN50→Res-18) | $85.72_{\pm 0.96}$ | $\mathbf{74.15}_{\pm 2.54}$ | $\mathbf{6.2}_{\pm 1.73}$ | $\mathbf{9.83}_{\pm 2.77}$ |
| FWD (Res-18→Res-18) | $84.3_{\pm 0.34}$ | $71.43_{\pm 1.8}$ | $19.33_{\pm 0.86}$ | $21.67_{\pm 0.53}$ |
| BIRD (Res-18→Res-18) | $\mathbf{84.49}_{\pm 0.18}$ | $\mathbf{77.04}_{\pm 0.35}$ | $\mathbf{11.32}_{\pm 1.29}$ | $\mathbf{16.15}_{\pm 1.38}$ |
| FWD (Shuffv2→Shuffv2) | $85.42_{\pm 0.42}$ | $70.13_{\pm 8.12}$ | $20.59_{\pm 2.22}$ | $27.2_{\pm 1.68}$ |
| BIRD (Shuffv2→Shuffv2) | $\mathbf{88.01}_{\pm 0.27}$ | $\mathbf{79.82}_{\pm 0.29}$ | $\mathbf{5.01}_{\pm 1.04}$ | $\mathbf{8.16}_{\pm 2.17}$ |
| FWD (CLIP-ViT-32→CLIP-ViT-32) | $87.16_{\pm 0.27}$ | $78.14_{\pm 0.41}$ | $22.87_{\pm 1.43}$ | $24.47_{\pm 1.61}$ |
| BIRD (CLIP-ViT-32→CLIP-ViT-32) | $\mathbf{88.55}_{\pm 0.03}$ | $\mathbf{80.84}_{\pm 0.06}$ | $\mathbf{3.44}_{\pm 0.92}$ | $\mathbf{5.19}_{\pm 1.06}$ |
| FWD (CLIP-RN50→CLIP-RN50) | $87.54_{\pm 0.14}$ | $78.87_{\pm 0.22}$ | $21.79_{\pm 0.68}$ | $22.88_{\pm 1.0}$ |
| BIRD (CLIP-RN50→CLIP-RN50) | $\mathbf{87.93}_{\pm 0.01}$ | $\mathbf{80.34}_{\pm 0.08}$ | $\mathbf{2.65}_{\pm 0.29}$ | $\mathbf{4.49}_{\pm 0.48}$ |
| FWD (Flava→Flava) | $84.51_{\pm 0.1}$ | $75.08_{\pm 0.5}$ | $26.96_{\pm 0.54}$ | $28.67_{\pm 1.18}$ |
| BIRD (Flava→Flava) | $\mathbf{85.48}_{\pm 0.02}$ | $\mathbf{78.54}_{\pm 2.1}$ | $\mathbf{2.53}_{\pm 0.17}$ | $\mathbf{4.12}_{\pm 0.59}$ |

Table 7: Results of BIRD compared to FWD (Chai et al., 2022) on UTK dataset for several architectures. BIRD consistently outperforms the additional baseline on fairness metrics and is competitive in terms of predictive performance for most cases.

| Method | AUROC (↑) | F1 (↑) | $\Delta_{\text{mean-DEO}}$ | $\Delta_{\text{max-DEO}}$ |
|---|---|---|---|---|
| FWD (CLIP-ViT-32→Res-18) | $95.12_{\pm 0.14}$ | $81.74_{\pm 0.35}$ | $16.19_{\pm 0.67}$ | $35.62_{\pm 1.86}$ |
| BIRD (CLIP-ViT-32→Res-18) | $91.05_{\pm 0.15}$ | $77.71_{\pm 0.61}$ | $\mathbf{14.96}_{\pm 1.23}$ | $\mathbf{30.55}_{\pm 2.68}$ |
| FWD (CLIP-RN50→Res-18) | $\mathbf{95.13}_{\pm 0.05}$ | $\mathbf{81.69}_{\pm 0.43}$ | $16.85_{\pm 0.31}$ | $35.12_{\pm 0.86}$ |
| BIRD (CLIP-RN50→Res-18) | $90.59_{\pm 0.25}$ | $77.04_{\pm 0.87}$ | $17.84_{\pm 1.3}$ | $\mathbf{33.53}_{\pm 1.21}$ |
| FWD (Res-18→Res-18) | $92.36_{\pm 0.29}$ | $78.63_{\pm 0.74}$ | $18.18_{\pm 1.12}$ | $35.62_{\pm 1.98}$ |
| BIRD (Res-18→Res-18) | $\mathbf{91.67}_{\pm 0.29}$ | $\mathbf{77.71}_{\pm 0.44}$ | $\mathbf{15.49}_{\pm 0.95}$ | $\mathbf{30.65}_{\pm 2.69}$ |
| FWD (Shuffv2→Shuffv2) | $89.36_{\pm 0.41}$ | $73.43_{\pm 0.79}$ | $23.18_{\pm 0.94}$ | $41.49_{\pm 1.8}$ |
| BIRD (Shuffv2→Shuffv2) | $\mathbf{89.53}_{\pm 0.37}$ | $\mathbf{73.88}_{\pm 0.61}$ | $\mathbf{16.92}_{\pm 1.15}$ | $\mathbf{36.12}_{\pm 1.39}$ |
| FWD (CLIP-ViT-32→CLIP-ViT-32) | $96.03_{\pm 0.07}$ | $86.34_{\pm 0.18}$ | $13.20_{\pm 0.63}$ | $23.68_{\pm 2.16}$ |
| BIRD (CLIP-ViT-32→CLIP-ViT-32) | $95.5_{\pm 0.02}$ | $85.67_{\pm 0.28}$ | $\mathbf{12.07}_{\pm 0.5}$ | $\mathbf{16.92}_{\pm 1.39}$ |
| FWD (CLIP-RN50→CLIP-RN50) | $95.68_{\pm 0.02}$ | $85.25_{\pm 0.13}$ | $13.70_{\pm 0.14}$ | $24.18_{\pm 0.43}$ |
| BIRD (CLIP-RN50→CLIP-RN50) | $95.43_{\pm 0.02}$ | $84.05_{\pm 0.13}$ | $\mathbf{14.43}_{\pm 0.14}$ | $\mathbf{23.28}_{\pm 0.43}$ |
| FWD (Flava→Flava) | $94.51_{\pm 0.04}$ | $81.27_{\pm 0.26}$ | $14.66_{\pm 0.68}$ | $32.54_{\pm 1.68}$ |
| BIRD (Flava→Flava) | $\mathbf{94.00}_{\pm 0.02}$ | $80.47_{\pm 0.54}$ | $\mathbf{14.23}_{\pm 0.55}$ | $\mathbf{28.16}_{\pm 2.89}$ |

Table 8: Results of Base KD on different teacher-student pairs (Q1). Shown is the average performance across five independent runs. We establish that across different architectures, KD results in unfair student models by following the fairness properties ($\Delta_{\text{mean-DEO}}$, $\Delta_{\text{max-DEO}}$) of the teacher.

| Baselines | | AUROC ($\uparrow$) | F1-score ($\uparrow$) | $\Delta_{\text{mean-DEO}}(\downarrow)$ | $\Delta_{\text{max-DEO}}(\downarrow)$ |
|---|---|---|---|---|---|
| FLAVA | | $84.43_{\pm0.12}$ | $74.87_{\pm0.63}$ | $27.48_{\pm0.64}$ | $29.37_{\pm1.53}$ |
| CLIP-ViT-32 | | $87.01_{\pm0.26}$ | $78.15_{\pm0.52}$ | $23.38_{\pm1.72}$ | $24.91_{\pm1.15}$ |
| CLIP-R50 | | $87.72_{\pm0.06}$ | $78.71_{\pm0.21}$ | $21.11_{\pm0.30}$ | $21.97_{\pm0.40}$ |
| ResNet18 | | $75.52_{\pm0.70}$ | $74.26_{\pm1.59}$ | $18.60_{\pm1.46}$ | $21.46_{\pm1.61}$ |
| ResNet34 | | $85.93_{\pm0.31}$ | $75.73_{\pm1.25}$ | $20.43_{\pm1.11}$ | $24.71_{\pm1.71}$ |
| Teacher | Student | AUROC ($\uparrow$) | F1-score ($\uparrow$) | $\Delta_{\text{mean-DEO}}(\downarrow)$ | $\Delta_{\text{max-DEO}}(\downarrow)$ |
| FLAVA | ResNet18 | $85.24_{\pm0.09}$ | $76.30_{\pm0.81}$ | $28.15_{\pm0.75}$ | $32.74_{\pm1.65}$ |
| FLAVA | ResNet34 | $84.81_{\pm0.32}$ | $75.46_{\pm0.89}$ | $29.50_{\pm1.29}$ | $33.34_{\pm2.29}$ |
| CLIP-ViT-32 | ResNet18 | $86.94_{\pm0.37}$ | $77.14_{\pm1.55}$ | $24.75_{\pm1.46}$ | $30.01_{\pm1.65}$ |
| CLIP-ViT-32 | ResNet34 | $86.71_{\pm0.15}$ | $78.11_{\pm0.39}$ | $23.68_{\pm0.24}$ | $25.38_{\pm1.00}$ |
| CLIP-R50 | ResNet18 | $86.71_{\pm0.37}$ | $77.96_{\pm0.63}$ | $25.61_{\pm1.86}$ | $30.08_{\pm1.49}$ |
| CLIP-R50 | ResNet34 | $87.31_{\pm0.15}$ | $78.11_{\pm0.39}$ | $23.68_{\pm0.24}$ | $25.38_{\pm1.0}$ |
| ResNet34 | ResNet18 | $86.25_{\pm0.76}$ | $73.45_{\pm1.30}$ | $19.80_{\pm1.62}$ | $23.86_{\pm1.67}$ |

Table 9: Hyperparameters for BIRD for different datasets and architectures. We perform minimal linear probing to find the optimal $\lambda$ (See 4.2) for each setting.

| Architecture | Dataset | $\lambda$ | Warmup | Total Epochs |
|---|---|---|---|---|
| FLAVA | CelebA | 0.1 | 5 | 10 |
| CLIP-ViT-32 | CelebA | 0.2 | 5 | 10 |
| CLIP-R50 | CelebA | 0.1 | 5 | 10 |
| ShuffleNetV2 | CelebA | 0.05 | 5 | 10 |
| ResNet18 | CelebA | 0.2 | 5 | 10 |
| ResNet34 | CelebA | 0.1 | 5 | 10 |
| FLAVA | UTK | 0.1 | 20 | 50 |
| CLIP-ViT-32 | UTK | 0.2 | 20 | 50 |
| CLIP-R50 | UTK | 0.1 | 20 | 50 |
| ShuffleNetV2 | UTK | 0.05 | 20 | 50 |
| ResNet18 | UTK | 0.2 | 20 | 50 |
| ResNet34 | UTK | 0.1 | 20 | 50 |
| ResNet18 | CIFAR-10S | 0.2 | 70 | 100 |

Table 10: Comparison of BIRD with and without Augmentations on the UTKFace Dataset. BIRD with Augmentations, on an average across models, improves the predictive power (AUROC and F1-Score) over BIRD while keeping the fairness criterion ($\Delta_{\text{mean-DEO}}$ and $\Delta_{\text{max-DEO}}$) same as w/o Augmentation.

| Teacher | Student | Method | AUROC ($\uparrow$) | F1-score ($\uparrow$) | $\Delta_{\text{mean-DEO}}(\downarrow)$ | $\Delta_{\text{max-DEO}}(\downarrow)$ |
|---|---|---|---|---|---|---|
| CLIP/RN50 | ResNet18 | BIRD | $90.59_{\pm0.13}$ | $77.04_{\pm0.87}$ | $17.84_{\pm1.30}$ | $33.35_{\pm1.21}$ |
| CLIP/RN50 | ResNet18 | BIRD + Aug | $\mathbf{91.05}_{\pm0.14}$ | $\mathbf{78.26}_{\pm0.72}$ | $\mathbf{16.00}_{\pm1.35}$ | $\mathbf{32.84}_{\pm1.70}$ |
| CLIP/ViT-32 | ResNet18 | BIRD | $\mathbf{91.05}_{\pm0.15}$ | $77.71_{\pm0.61}$ | $\mathbf{14.96}_{\pm1.23}$ | $\mathbf{30.55}_{\pm2.68}$ |
| CLIP/ViT-32 | ResNet18 | BIRD + Aug | $90.93_{\pm0.28}$ | $\mathbf{77.74}_{\pm0.63}$ | $16.48_{\pm0.70}$ | $34.73_{\pm1.54}$ |
| CLIP/RN50 | CLIP/RN50 | BIRD | $95.43_{\pm0.02}$ | $84.05_{\pm0.13}$ | $\mathbf{12.43}_{\pm0.14}$ | $23.28_{\pm0.43}$ |
| CLIP/RN50 | CLIP/RN50 | BIRD + Aug | $\mathbf{95.93}_{\pm0.02}$ | $\mathbf{85.27}_{\pm0.29}$ | $12.74_{\pm0.51}$ | $\mathbf{18.11}_{\pm0.30}$ |
| CLIP/ViT-32 | CLIP/ViT-32 | BIRD | $95.50_{\pm0.04}$ | $85.67_{\pm0.08}$ | $\mathbf{12.07}_{\pm0.27}$ | $\mathbf{16.92}_{\pm0.82}$ |
| CLIP/ViT-32 | CLIP/ViT-32 | BIRD + Aug | $\mathbf{96.03}_{\pm0.03}$ | $\mathbf{87.04}_{\pm0.24}$ | $12.14_{\pm0.4}$ | $19.00_{\pm1.45}$ |

Table 11: Results of BIRD against baselines for CLIP/ViT-32 and CLIP/RN50 as students on the CelebA dataset. BIRD retains the predictive power (AUROC and F1-Score) of the baseline model while improving the fairness ($\Delta_{\text{mean-DEO}}$ and $\Delta_{\text{max-DEO}}$) while traditional KD methods perform poorly. (Note that the optimal value of $\alpha$ in this setting was found to be 0.1 based on linear probing.)

| Teacher | Student | Method | AUROC (↑) | F1-score (↑) | $\Delta_{\text{mean-DEO}}$(↓) | $\Delta_{\text{max-DEO}}$(↓) |
|---|---|---|---|---|---|---|
| ResNet18 | CLIP/ViT-32 | Baseline | 85.44±0.29 | 74.26±1.59 | 18.60±1.46 | 21.46±1.61 |
| ResNet18 | CLIP/ViT-32 | BKD | 86.94±0.06 | 69.62±1.56 | 20.98±0.64 | 29.26±0.77 |
| ResNet18 | CLIP/ViT-32 | FitNet | 87.12±0.17 | 69.88±1.61 | 20.64±0.58 | 28.76±1.39 |
| ResNet18 | CLIP/ViT-32 | AD | **88.91**±0.07 | 74.23±0.84 | **6.35**±1.29 | **8.01**±1.87 |
| ResNet18 | CLIP/ViT-32 | MFD | 84.03±1.02 | 74.25±1.01 | 19.54±4.73 | 24.64±6.47 |
| ResNet18 | CLIP/ViT-32 | BIRD | 88.55±0.00 | **79.51**±0.59 | 6.73±1.51 | 8.87±1.88 |
| ResNet18 | CLIP/RN50 | Baseline | 85.44±0.29 | 74.26±1.59 | 18.60±1.46 | 21.46±1.61 |
| ResNet18 | CLIP/RN50 | BKD | 87.30±0.06 | 71.94±0.80 | 18.81±0.65 | 26.30±1.23 |
| ResNet18 | CLIP/RN50 | FitNet | 87.25±0.12 | 71.17±1.18 | 19.09±1.10 | 27.03±1.89 |
| ResNet18 | CLIP/RN50 | AD | 88.24±0.03 | 70.61±0.86 | 4.95±0.77 | 7.39±0.96 |
| ResNet18 | CLIP/RN50 | MFD | 86.70±0.36 | 77.50±0.79 | 20.81±1.50 | 23.86±1.06 |
| ResNet18 | CLIP/RN50 | BIRD | **89.13**±0.05 | **80.53**±0.44 | **3.47**±0.78 | **4.88**±1.2 |
| ResNet34 | CLIP/ViT-32 | Baseline | 85.93±0.31 | 75.46±1.25 | 20.43±1.11 | 24.71±1.71 |
| ResNet34 | CLIP/ViT-32 | BKD | 86.85±0.09 | 70.04±0.9 | 21.38±0.64 | 29.35±1.06 |
| ResNet34 | CLIP/ViT-32 | FitNet | 86.99±0.14 | 71.12±1.11 | 21.64±0.46 | 29.28±1.38 |
| ResNet34 | CLIP/ViT-32 | AD | **88.97**±0.08 | 73.35±1.45 | 7.08±0.75 | 9.45±1.15 |
| ResNet34 | CLIP/ViT-32 | MFD | 83.45±0.60 | 73.57±1.84 | 23.96±2.78 | 30.72±5.12 |
| ResNet34 | CLIP/ViT-32 | BIRD | 88.67±0.09 | **79.79**±0.61 | **4.55**±0.65 | **6.25**±1.01 |
| ResNet34 | CLIP/RN50 | Baseline | 85.93±0.31 | 75.73±1.25 | 20.43±1.11 | 24.71±1.71 |
| ResNet34 | CLIP/RN50 | BKD | 87.52±0.22 | 71.30±0.98 | 17.95±1.29 | 25.37±1.92 |
| ResNet34 | CLIP/RN50 | FitNet | 87.24±0.09 | 72.59±0.98 | 19.95±0.64 | 27.11±0.77 |
| ResNet34 | CLIP/RN50 | AD | 88.25±0.03 | 70.94±1.64 | 5.01±0.80 | 7.21±0.73 |
| ResNet34 | CLIP/RN50 | MFD | 86.90±0.27 | 75.83±1.66 | 18.46±1.20 | 22.5±0.53 |
| ResNet34 | CLIP/RN50 | BIRD | **89.05**±0.1 | **80.95**±0.23 | **3.66**±1.4 | **5.63**±1.75 |

Table 12: Results of KD using foundation models, ResNets, and two fairness datasets. Shown is the average performance across five independent runs. Arrows (↑, ↓) indicate the direction of better performance. BIRD retains the predictive power (Top1, F1-score) of the baseline model while improving their fairness (shaded area).

| Model | Dataset | Method | Top1 (↑) | F1-score (↑) | $\Delta_{\text{mean-DEO}}$(↓) | $\Delta_{\text{max-DEO}}$(↓) |
|---|---|---|---|---|---|---|
| CLIP-ViT-32 →ResNet18 | CelebA | Student | 75.52±0.70 | 74.26±1.59 | 18.60±1.46 | 21.46±1.61 |
| | | Teacher | 77.83±0.30 | 78.15±0.52 | 23.38±0.20 | 24.91±1.15 |
| | | BKD | 77.55±0.43 | 77.55±0.94 | 25.04±1.46 | 28.81±1.53 |
| | | FitNet | 78.31±0.31 | 78.85±0.52 | 23.96±1.75 | 26.29±1.38 |
| | | AD | 67.71±3.81 | 71.61±1.81 | 8.48±3.55 | 11.30±3.21 |
| | | MFD | 77.86±0.28 | 77.86±0.28 | 18.31±1.60 | 25.27±0.84 |
| | | **BIRD** | 80.51±0.12 | 80.86±0.15 | **7.51**±1.03 | **10.16**±1.12 |
| | UTKFace | Student | 78.73±0.27 | 78.73±0.27 | 17.21±0.40 | 36.92±1.13 |
| | | BKD | 81.12±0.56 | 81.12±0.56 | 17.74±0.82 | 36.22±0.82 |
| | | FitNet | 81.59±0.52 | 81.59±0.52 | 16.52±0.69 | 33.63±1.81 |
| | | AD | 69.03±2.12 | 69.03±0.57 | 30.55±2.19 | 50.85±4.55 |
| | | MFD | 77.61±0.25 | 77.61±0.25 | 18.31±1.60 | 34.73±2.21 |
| | | **BIRD** | 77.71±0.61 | 77.71±0.61 | **14.96**±1.23 | **30.55**±2.68 |
| CLIP-R50 →ResNet18 | CelebA | Student | 75.52±0.70 | 74.26±1.59 | 18.60±1.46 | 21.46±1.61 |
| | | BKD | 78.35±0.19 | 78.72±0.47 | 22.99±0.47 | 26.25±1.39 |
| | | FitNet | 78.22±0.12 | 81.59±0.52 | 22.93±0.32 | 26.51±0.8 |
| | | AD | 68.79±1.19 | 62.85±3.21 | 17.80±6.43 | 25.52±9.54 |
| | | MFD | 77.80±0.19 | 77.55±0.37 | 21.88±0.38 | 23.52±0.65 |
| | | **BIRD** | 75.24±1.75 | 74.15±2.54 | **6.20**±1.73 | **9.83**±2.77 |
| | UTKFace | Student | 78.73±0.27 | 78.73±0.27 | 17.21±0.40 | 36.92±1.13 |
| | | BKD | 82.51±0.62 | 82.51±0.62 | 16.85±0.86 | 35.52±1.07 |
| | | FitNet | 81.42±0.38 | 81.42±0.38 | **16.25**±1.08 | 34.33±2.11 |
| | | AD | 68.28±1.88 | 68.28±1.88 | 28.46±1.46 | 53.33±2.72 |
| | | MFD | 75.87±1.10 | 75.87±1.10 | 18.77±1.07 | 39.90±1.99 |
| | | **BIRD** | 77.26±0.27 | 77.04±0.87 | 17.84±1.30 | **33.35**±1.21 |

Table 13: Results of self-distillation on foundation models using two fairness datasets. Shown is the average performance across five independent runs. Arrows ($\uparrow$, $\downarrow$) indicate the direction of better performance. BIRD retains the predictive power (Top1-acc and F1-score) of the baseline model while improving their fairness (shaded area). Note that teacher-student architectures in each case are identical.

| Model | Dataset | Method | Top1 ($\uparrow$) | F1-score ($\uparrow$) | $\Delta_{\text{mean-DEO}}(\downarrow)$ | $\Delta_{\text{max-DEO}}(\downarrow)$ |
|---|---|---|---|---|---|---|
| CLIP-ViT-32 | UTKFace | Baseline | $86.22_{\pm0.16}$ | $86.22_{\pm0.16}$ | $13.47_{\pm0.20}$ | $25.07_{\pm0.96}$ |
| | | BKD | $86.02_{\pm0.15}$ | $86.02_{\pm0.15}$ | $13.73_{\pm0.15}$ | $25.27_{\pm0.99}$ |
| | | FitNet | $86.27_{\pm0.31}$ | $86.07_{\pm0.29}$ | $13.80_{\pm0.21}$ | $25.47_{\pm1.42}$ |
| | | AD | $86.34_{\pm0.32}$ | $86.34_{\pm0.32}$ | $\mathbf{11.84}_{\pm0.73}$ | $22.49_{\pm0.31}$ |
| | | MFD | $86.64_{\pm0.12}$ | $86.64_{\pm0.15}$ | $12.11_{\pm0.16}$ | $22.79_{\pm0.37}$ |
| | | BIRD | $85.67_{\pm0.08}$ | $85.67_{\pm0.08}$ | $12.07_{\pm0.27}$ | $\mathbf{16.92}_{\pm0.82}$ |
| | CelebA | Baseline | $77.83_{\pm0.30}$ | $78.15_{\pm0.52}$ | $23.38_{\pm1.72}$ | $24.91_{\pm1.15}$ |
| | | BKD | $77.88_{\pm0.26}$ | $78.20_{\pm0.48}$ | $23.26_{\pm1.67}$ | $24.62_{\pm1.14}$ |
| | | FitNet | $77.97_{\pm0.14}$ | $78.16_{\pm0.46}$ | $23.25_{\pm1.50}$ | $24.57_{\pm1.02}$ |
| | | AD | $78.76_{\pm0.15}$ | $79.00_{\pm0.15}$ | $17.02_{\pm1.03}$ | $17.82_{\pm0.97}$ |
| | | MFD | $77.84_{\pm0.28}$ | $77.59_{\pm0.70}$ | $21.99_{\pm0.70}$ | $23.70_{\pm1.58}$ |
| | | BIRD | $80.13_{\pm0.07}$ | $80.84_{\pm0.06}$ | $\mathbf{3.44}_{\pm0.92}$ | $\mathbf{5.19}_{\pm1.06}$ |
| CLIP-ResNet50 | UTKFace | Baseline | $84.9_{\pm0.2}$ | $84.90_{\pm0.20}$ | $13.70_{\pm0.58}$ | $23.08_{\pm1.06}$ |
| | | BKD | $84.85_{\pm0.2}$ | $84.85_{\pm0.20}$ | $13.57_{\pm0.50}$ | $23.28_{\pm0.99}$ |
| | | FitNet | $85.02_{\pm0.44}$ | $84.98_{\pm0.28}$ | $13.93_{\pm0.45}$ | $23.78_{\pm1.01}$ |
| | | AD | $83.86_{\pm0.16}$ | $83.86_{\pm0.16}$ | $14.83_{\pm1.58}$ | $26.27_{\pm0.70}$ |
| | | MFD | $84.90_{\pm0.52}$ | $84.90_{\pm0.52}$ | $14.16_{\pm0.60}$ | $\mathbf{22.99}_{\pm1.60}$ |
| | | BIRD | $84.05_{\pm0.13}$ | $84.05_{\pm0.13}$ | $\mathbf{12.43}_{\pm0.14}$ | $23.28_{\pm0.43}$ |
| | CelebA | Baseline | $78.61_{\pm0.10}$ | $78.71_{\pm0.21}$ | $21.11_{\pm0.30}$ | $21.97_{\pm0.41}$ |
| | | BKD | $78.64_{\pm0.03}$ | $78.90_{\pm0.15}$ | $21.10_{\pm0.40}$ | $22.07_{\pm0.41}$ |
| | | FitNet | $78.33_{\pm0.17}$ | $78.77_{\pm0.21}$ | $21.00_{\pm0.28}$ | $21.80_{\pm0.38}$ |
| | | AD | $80.07_{\pm0.03}$ | $80.32_{\pm0.05}$ | $5.33_{\pm0.19}$ | $7.93_{\pm0.22}$ |
| | | MFD | $78.28_{\pm0.12}$ | $78.56_{\pm0.21}$ | $22.56_{\pm0.56}$ | $23.52_{\pm0.33}$ |
| | | BIRD | $80.04_{\pm0.05}$ | $80.34_{\pm0.08}$ | $\mathbf{2.65}_{\pm0.29}$ | $\mathbf{4.49}_{\pm0.48}$ |

Table 14: Results of BIRD against baselines for UTKFace Dataset with the same augmentations applied to each technique.

| Teacher | Student | Method | AUROC ($\uparrow$) | F1-score ($\uparrow$) | $\Delta_{\text{mean-DEO}}(\downarrow)$ | $\Delta_{\text{max-DEO}}(\downarrow)$ |
|---|---|---|---|---|---|---|
| CLIP/ViT-32 | CLIP/ViT-32 | BKD | $95.96_{\pm0.06}$ | $85.97_{\pm0.34}$ | $13.96_{\pm0.18}$ | $25.57_{\pm1.72}$ |
| CLIP/ViT-32 | CLIP/ViT-32 | FitNet | $95.95_{\pm0.06}$ | $86.07_{\pm0.29}$ | $13.80_{\pm0.21}$ | $25.47_{\pm1.42}$ |
| CLIP/ViT-32 | CLIP/ViT-32 | AD | $96.08_{\pm0.06}$ | $86.17_{\pm0.22}$ | $13.27_{\pm0.21}$ | $23.98_{\pm1.16}$ |
| CLIP/ViT-32 | CLIP/ViT-32 | MFD | $96.05_{\pm0.04}$ | $86.64_{\pm0.15}$ | $\mathbf{12.11}_{\pm0.16}$ | $22.79_{\pm0.37}$ |
| CLIP/ViT-32 | CLIP/ViT-32 | BIRD | $\mathbf{96.03}_{\pm0.03}$ | $\mathbf{87.04}_{\pm0.24}$ | $12.14_{\pm0.40}$ | $\mathbf{19.00}_{\pm1.45}$ |
| CLIP/RN50 | CLIP/RN50 | BKD | $95.66_{\pm0.04}$ | $84.93_{\pm0.21}$ | $13.60_{\pm0.50}$ | $23.58_{\pm1.05}$ |
| CLIP/RN50 | CLIP/RN50 | FitNet | $95.66_{\pm0.03}$ | $84.95_{\pm0.29}$ | $13.96_{\pm0.47}$ | $23.58_{\pm1.01}$ |
| CLIP/RN50 | CLIP/RN50 | AD | $95.66_{\pm0.05}$ | $83.71_{\pm0.15}$ | $15.46_{\pm0.44}$ | $29.05_{\pm0.93}$ |
| CLIP/RN50 | CLIP/RN50 | MFD | $95.68_{\pm0.03}$ | $84.88_{\pm0.51}$ | $14.20_{\pm0.59}$ | $22.99_{\pm1.60}$ |
| CLIP/RN50 | CLIP/RN50 | BIRD | $\mathbf{95.93}_{\pm0.02}$ | $\mathbf{85.27}_{\pm0.29}$ | $\mathbf{12.74}_{\pm0.51}$ | $\mathbf{18.11}_{\pm0.30}$ |
| CLIP/ViT-32 | ResNet18 | BKD | $\mathbf{94.9}_{\pm0.05}$ | $81.17_{\pm0.44}$ | $17.94_{\pm0.92}$ | $38.31_{\pm2.25}$ |
| CLIP/ViT-32 | ResNet18 | FitNet | $94.70_{\pm0.12}$ | $\mathbf{81.99}_{\pm0.52}$ | $\mathbf{15.32}_{\pm0.57}$ | $\mathbf{33.73}_{\pm1.06}$ |
| CLIP/ViT-32 | ResNet18 | AD | $93.56_{\pm0.15}$ | $79.75_{\pm0.35}$ | $18.91_{\pm1.25}$ | $38.71_{\pm3.65}$ |
| CLIP/ViT-32 | ResNet18 | MFD | $90.23_{\pm0.29}$ | $77.21_{\pm0.36}$ | $17.31_{\pm1.26}$ | $35.72_{\pm2.50}$ |
| CLIP/ViT-32 | ResNet18 | BIRD | $90.78_{\pm0.28}$ | $77.36_{\pm0.64}$ | $16.64_{\pm0.60}$ | $34.91_{\pm1.27}$ |
| CLIP/RN50 | ResNet18 | BKD | $\mathbf{95.29}_{\pm0.14}$ | $\mathbf{82.11}_{\pm0.40}$ | $16.52_{\pm0.50}$ | $35.52_{\pm1.62}$ |
| CLIP/RN50 | ResNet18 | FitNet | $94.86_{\pm0.05}$ | $81.52_{\pm0.32}$ | $17.15_{\pm0.60}$ | $37.71_{\pm1.38}$ |
| CLIP/RN50 | ResNet18 | AD | $93.38_{\pm0.21}$ | $78.61_{\pm0.56}$ | $21.03_{\pm0.96}$ | $44.88_{\pm5.39}$ |
| CLIP/RN50 | ResNet18 | MFD | $90.64_{\pm0.25}$ | $77.54_{\pm0.36}$ | $17.61_{\pm0.91}$ | $35.92_{\pm2.32}$ |
| CLIP/RN50 | ResNet18 | BIRD | $91.14_{\pm0.15}$ | $77.86_{\pm0.85}$ | $\mathbf{16.09}_{\pm1.91}$ | $\mathbf{33.33}_{\pm2.30}$ |

