# OpenReview forum: "Towards Fair Knowledge Distillation using Student Feedback"
_ICLR.cc/2024/Conference — Submitted to ICLR 2024_

### Official Review · Reviewer_QvRs · 2023-10-24

**Soundness:** 3 good
**Presentation:** 3 good
**Contribution:** 3 good
**Rating:** 6
**Confidence:** 3

**Summary:**

This paper targets on the fairness problem in knowledge distillation. The proposed method, BIRD, collects the feedback from the student model through a meta-learning-based approach and selectively distill teacher knowledge. BIRD is orthogonal with existing methods and computationally effective. Extensive experiment results show that BIRD can enhance the fairness remarkably.

**Strengths:**

1. This paper is overall well-written and easy to follow.
2. This paper targets on an interesting problem, the fairness in knowledge distillation.
3. The proposed method is orthogonal with existing methods, enhancing the fairness remarkably.

**Weaknesses:**

1. I am curious about the results when we apply BIRD to existing methods, and test their performance according to the conventional criteria? eg. The Top-1 and Top-5 accuracy on CIFAR-100 and ImageNet. Will it degrade when pursuing fairness? I want to see more comparison results.

**Questions:**

See weakness.

---

> ### Author Response · Authors · 2023-11-17
> **Rebuttal response to Reviewer QvRs**
>
> We are very excited to learn that the reviewer finds our work well motivated/presented, our method novel as compared to existing works, and our results remarkably enhancing fairness. We appreciate your helpful feedback and provide further details to answer your question below.
>
> **Including more conventional metrics**
>
> We thank the reviewer for this valuable suggestion and agree that the inclusion of the aforementioned criteria will account for a more thorough representation of the accuracy results in our work. We would like to note that the number of classes in our benchmark datasets is 2 (for CelebA) and 4 (for UTKFace), respectively. Following the reviewers’ feedback, we report the top-1 accuracies for the experiments in Table 1 and Table 2 in the Appendix of the updated manuscript (**see Table 12-13**). We are excited to share that our results follow the same trend and BIRD improves the fairness of the student model without sacrificing its predictive performance. Further, as explained below we cannot calculate metrics like top-5 accuracy in our experiments due to the limited number of classes in the fairness datasets.
>
> We have included this clarification in the revised manuscript and hope we have addressed all your questions adequately. In light of these clarifications, we kindly request you consider increasing your score. We are happy to answer any further questions.

---

### Official Review · Reviewer_qpK3 · 2023-10-31

**Soundness:** 3 good
**Presentation:** 3 good
**Contribution:** 3 good
**Rating:** 6
**Confidence:** 3

**Summary:**

This study introduces a "student-aware selective feature distillation" approach, drawing inspiration from meta-learning, which enables the teacher to impart unbiased information effectively to the student.

**Strengths:**

1. The authors have lucidly articulated both the problem definition and the corresponding solution formulation.

2. Distinguishing itself from prior research, this study emphasizes adapting the teacher's predictions to remain unbiased. This adjustment, in turn, facilitates the training of a student that naturally inherits this unbiased character due to the teacher's modified features.

3. The introduction of a meta-learning inspired transfer approach is both well-conceived and aptly presented, offering a compelling solution for unbiased (or fair) knowledge transfer.

4. In the results section, the authors comprehensively address pivotal concerns, including the problem's justification (5.2.1), the efficacy of their proposed method (5.2.2), the technique's adaptability across various KD frameworks (5.2.3), and insightful ablation studies that dissect various components of their framework (5.2.4 and 5.2.5).

**Weaknesses:**

1. While the paper's overarching framework appears to draw heavily from the "learning how to teach" paradigm (as detailed in Park et al., 2021; Liu et al., 2021; Zhou et al., 2021), its overall contribution may be perceived as somewhat incremental. This perception arises from the adaptation of a pre-established framework to address knowledge distillation. Despite this, the proposed solution stands out due to its technical novelty and apt alignment with the problem statement. A deeper dive by the authors into the technical distinctions between their work and prior studies would further solidify their contribution.

2. For the benefit of practitioners, the authors might consider expanding their ablation section to detail not just the memory overhead, but also the time overhead associated with the distillation process. This is particularly relevant given the well-documented time-intensive nature of meta-learning-based approaches.

**Questions:**

Can the authors detail how the CLIP-Resnet-50 to ResNet 18 KD is performed. How is the KD distillation performed in the absence of the class-probability distribution of CLIP?

---

> ### Author Response · Authors · 2023-11-17
> **Rebuttal response to Reviewer qpK3**
>
> We thank the reviewer for acknowledging the proposed framework as compelling and our results comprehensive. We greatly appreciate your feedback on solidifying the technical contributions of our work with the related works and extending our ablation studies.
>
> **Learning how to teach paradigm and technical differences with related works**
>
> We thank the reviewer for recognizing the *technical novelty* and *alignment* with the problem statement. While we agree with the reviewer that the referenced works utilize some variant of learning how to teach framework, they focus on fine-tuning/re-training the teacher model, **which is not the goal of our work** as we assume black-box access of the teacher and only use its output representations. Below we provide a deeper dive into the technical distinctions between the referenced works and Bird.
>
> *Liu Et.al 2021: Learning to teach with student feedback*
>
> The paper proposes IKD to establish a bidirectional interaction between the teacher and the student models. **Unlike traditional KD methods, where the teacher model is fixed** and provides static soft targets for the student model, IKD enables the teacher model to adapt its teaching strategy based on feedback from the student. Precisely, the **teacher model is trained to generate specific soft targets at each training step using meta-learning**, especially MAML (Finn, Abbeel, and Levine 2017). In contrast, our proposed framework, Bird, does not modify the weights of the original teacher model.
>
> *Park Et. al 2021: Learning Student-Friendly Teacher Networks for Knowledge Distillation (SFTN)*
>
> Their goal is to learn networks that would serve as better teachers to the students. To achieve this, they add **proxy student branches to the teacher model** during its training and then **optimize the teacher**, such that it minimizes the representation difference between the teacher and student branches. The trained teacher is consequently used for knowledge distillation to standard student models using Vanilla KD (Hinton et. al). Their overarching principle is to learn a model that is better teacher in general and not optimize Knowledge Transfer with respect to a specific student.  Again, they modify the teacher model and its learned weights, **which is non-trivial and computationally expensive for state-of-the-art foundation models.**
>
> *Zhou et al 2022: BERT Learns to Teach: Knowledge Distillation with Meta-Learning*
>
> They introduce MetaDistil, which combines knowledge distillation with meta-learning. Unlike traditional knowledge distillation frameworks where the teacher model is fixed, MetaDistil allows the teacher model to be trainable and adjust based on the student model’s performance using meta-learning (similar to Liu et al).
>
> One key distinction between the above methods and Bird is that we do not fine-tune or modify the weights of the teacher model and only use the student feedback to re-weigh **penultimate** teacher representations during distillation. As rightly pointed out by the reviewer, our approach is aligned and motivated but addresses an unexplored area of fairness in knowledge distillation. While existing works utilize the learning how-to-teach paradigm, **they do so primarily for optimizing the students' predictive performance**, which often comes at the cost of retraining the entire teacher model.
>
> **Time Ablation**
>
> We thank the reviewer for this insightful question and agree that performing a time overhead study in addition to the memory overhead will be very helpful. In response to the reviewer’s suggestion, we perform this and share the time taken per iteration by BIRD (*note that this number is computed by averaging the time for all iterations in an epoch*). As rightly pointed out by the reviewer, *meta-learning is usually time-intensive.* However, we would like to highlight that even then BIRD achieves competitive time per iteration in addition to the fairness performance gains it offers. Our analysis indicates the time complexity can be attributed to the library used for meta-learning (*higher library*), which can be further optimized and remains a part of our future work (as discussed in the limitation).
>
> |     **Teacher --> Student**     |   **MFD**   | **Base KD** |   **BIRD**  |
> |:-------------------------------:|:-----------:|:-----------:|:-----------:|
> | CLIP-Vit32 --> CLIP-Vit32       | 0.1192 s/it | 0.0986 s/it | 3.1507 s/it |
> | CLIP-ResNet50 --> CLIP-ResNet50 | 0.0985 s/it | 0.0762 s/it | 3.1124 s/it |
> | CLIP-Vit32 --> ResNet18         | 0.2585 s/it | 0.1346 s/it | 4.5481 s/it |
> | CLIP-ResNet50 --> ResNet18      | 0.2321 s/it | 0.1193 s/it | 4.3356 s/it |
>
> We are very grateful to the reviewer for all their suggestions, which have helped us in differentiating our work from existing works. We tried our best to address all the questions in our response. In light of these updates, we kindly request the reviewer to consider increasing the score. We are very happy to answer any further questions.

---

### Official Review · Reviewer_UAGV · 2023-10-31

**Soundness:** 2 fair
**Presentation:** 4 excellent
**Contribution:** 1 poor
**Rating:** 5
**Confidence:** 5

**Summary:**

This paper investigates making student models with less gender & racial bias in knowledge distillation. The meta-learning framework is used to achieve this goal, and the experiments are done on CelebA and UTK datasets.

**Strengths:**

The motivation of this paper is sound, fairness in KD is indeed an important topic in the era of large models. This paper is well-presented.

**Weaknesses:**

1. This paper proposes a meta-learning framework to solve the fairness issue in KD. I don't see why meta-learning can be used to resolve fairness here; the intuition and motivation are unclear.


2. The experiments are mainly conducted on CelebA and UTKFace, two small datasets. I think it is necessary to evaluate a larger benchmark, such as a dataset that a mixup of gender/racial images with other non-biased images, i.e., animals,  to verify the effectiveness of the proposed method on a larger scale setting. The same issue is on the choice of student models; more models should be evaluated, such as ViT (not as a teacher but as a student model).

3. It is disappointing that this paper is only focused on the fairness of classification. The application of classification is very limited and has been extensively explored over the past decades. I think it is essential to test KD fairness on more settings, such as multi-turn vqa. Otherwise, I suggest changing the title to a more specific topic, i.e. "Towards Fair Knowledge Distillation on Image Classification".

**Questions:**

See weakness. My main concerns are the following: 1) it is not clear why the meta-learning framework is effective for **fairness** (not the overall results.), 2) the experiments are insufficient regarding the size of the dataset, the model size of the students, and the tasks.

---

> ### Author Response · Authors · 2023-11-17
> **Rebuttal response to Reviewer UAGV (Part 1/2)**
>
> We thank the reviewer for acknowledging the motivation and presentation of our work. We appreciate your helpful feedback and address all the mentioned questions/concerns below.
>
> **The intuition behind a Meta-Learning framework**
>
> We would first like to thank the reviewer for appreciating the problem of fairness in knowledge distillation (KD). Further, we would like to highlight that none of the existing works in KD literature **adapt incoming biased teacher knowledge based on student feedback**, which can **lead to a fairer student model**. Here, the intuition is to identify and select teacher features that do not enhance the bias in the student model. We utilize a meta-learning framework as it provides **structure to the process of learning from feedback** and helps better optimization. This means, that when the incoming features from the teacher model are biased, it gets that feedback through the fairness properties of an inner-learner (a proxy/copy student model; Line 7: Algorithm 1), and it is subsequently updated (Line 8: Algorithm 1). The updated teacher model is then used to optimize the main student fairly. Thus, the teacher first uses the proxy student to understand the student’s optimization process, which it leverages for fair teaching.
>
> Further, we conducted an ablation on the importance of the meta-step (research question 4 in Sec. 5.2 of the original and revised draft) in our BIRD framework and showed that the meta-step allows us to update the parameters of FairDistill such that the biased features from the teacher do not impact the student. More specifically, results show that Bird with meta-gradients shows a **6.45% improvement in the fairness** over Bird without meta-gradients, providing empirical evidence that the meta-gradients improve the fairness of the student model.
>
> **Evaluation of More Student Models**
>
> We thank the reviewer for this experiment suggestion. Indeed an evaluation with more student model architecture choices would provide a deeper understanding of our approach. Further, we would like to point out to the reviewer that **Table 4** in our attached supplementary section includes the results of BIRD with students of different sizes: ResNet34, ShufflenetV2, and FLAVA. In addition, we perform the additional experiments suggested by the reviewer choosing CLIP/ViT-32 and CLIP/RN-50 as students and RN18, and RN34 as teachers on the CelebA dataset. Please find the results for the aforementioned ablation in **Table 11** of the updated draft. We note that BIRD comfortably outperforms all baselines by a large margin, proving its efficacy across model sizes.
>
> **Larger Datasets**
>
> The two main datasets – UTKFace and CelebA – we utilize in this work are very popular and are widely employed in fairness research to date. For instance, several recent works in fairness published at **NeurIPS, AAAI, ICCV, CVPR, ECCV, and UAI conferences in 2022-23** have employed these datasets both to evaluate the efficacy of newly proposed methods, as well as to study the behavior of existing methods [1-10]. In addition, previous work in knowledge distillation, like MFD [1] and FWD [2], also rely on these datasets. Given these past works, we follow suit and employ these datasets in our benchmarking efforts. We agree with the reviewer that the necessity of evaluating models on a larger benchmark, such as a dataset with a mixup of images with different attributes, to verify the effectiveness of fairness methods is certainly important, and we aim to explore them in our follow-up work as they become available.

---

> > ### Author Response · Authors · 2023-11-17
> > **Rebuttal response to Reviewer UAGV (Part 2/2)**
> >
> > **Ambiguous Title of the Paper**
> >
> > While we acknowledge the reviewers' feedback that KD has been extensively studied in classification, we would like to highlight that research on fairness for KD in a classification setting using **foundation models** is still in its nascent stages, where **there has been no distillation work** that analyzes the fairness issues of distillation using foundation models. We agree that experimenting with KD tasks considerably different from classification, such as Multi-Turn VQA or image retrieval, involving both text and image input datasets is crucial. However, we would like to kindly mention to the reviewer that in the interest of the discussion period, we look forward to exploring such an adaptation as an interesting future direction. We also apologize to the reviewer for the ambiguity arising from our current paper title and appreciate the suggestion for the new name.
> >
> > We will include all the above clarifications and details in the final version. We hope we addressed all your questions/concerns/comments adequately. In light of these clarifications, we kindly request you to consider increasing your score. We are very happy to answer any further questions.
> >
> > **References**
> >
> > [1] Jung, Sangwon, et al. "Fair feature distillation for visual recognition" In CVPR, 2021
> >
> > [2] Chai, Junyi, Taeuk Jang, and Xiaoqian Wang. "Fairness without demographics through knowledge distillation" In NeurIPS, 2022
> >
> > [3] Dooley, S., Wei, G. Z., Goldstein, T., & Dickerson, J. Robustness Disparities in Face Detection. NeurIPS, 2022
> >
> > [4] Xu, Y., Guo, W., & Wei, Z. Conformal risk control for ordinal classification. In UAI, 2023
> >
> > [5] Lin, X., Kim, S., & Joo, J. Fairgrape: Fairness-aware gradient pruning method for face attribute classification. In ECCV, 2022
> >
> > [6] Qraitem, M., Saenko, K., & Plummer, B. A. Bias Mimicking: A Simple Sampling Approach for Bias Mitigation. In CVPR, 2023
> >
> > [7] Zhu, J., Gu, L., Wu, X., Li, Z., Harada, T., & Zhu, Y. People taking photos that face never share: privacy protection and fairness enhancement from camera to user. In AAAI, 2023
> >
> > [8] Wu, T. H., Su, H. T., Chen, S. T., & Hsu, W. H. Fair Robust Active Learning by Joint Inconsistency. In ICCV, 2023
> >
> > [9] Teo, C. T., Abdollahzadeh, M., & Cheung, N. M. Fair generative models via transfer learning. In AAAI, 2023
> >
> > [10] Um, S., & Suh, C. A fair generative model using lecam divergence. In AAAI, 2023

---

> > > ### Author Response · Authors · 2023-11-20
> > > **Looking forward to your response**
> > >
> > > Dear reviewer,
> > >
> > > Thank you again for your thoughtful feedback. Following your suggestions, we included new results on additional student models and clarified the intuition behind a meta-learning framework and the use of the UTKFace and CelebA datasets. We would love to hear your thoughts on our response. Please let us know if there is anything else we can do to address your comments.

---

> ### Comment · Reviewer_UAGV · 2023-12-04
> **comment**
>
> After carefully reading the authors' responses and other reviewers' comments, I raised my rating to 5. However, I am still inclined towards rejection due to concerns regarding the scale of the datasets.
>
> I also agree with Reviewer qaWv that the evaluation metrics may be problematic. Also, the motivation that needs to use internal features from the teacher model doesn't make sense to me.

---

### Official Review · Reviewer_qaWv · 2023-11-02

**Soundness:** 3 good
**Presentation:** 3 good
**Contribution:** 2 fair
**Rating:** 5
**Confidence:** 4

**Summary:**

This paper proposes a new fairness-aware KD method, BIRD (BIas-awaRe Distillation), for distilling foundation models. The main idea is to use a proposed FAIRDISTILL operator to collect feedback from the student through a meta-learning-based approach and selectively distill teacher knowledge.  This method can be used with several existing base KD methods for improved performance. Extensive experiments across three fairness datasets show the efficacy of the proposed method over other counterparts.

**Strengths:**

1. As FMs prevail day by day, distilling FMs is more important as well. The fairness problem of the distilled model is also of interest. This paper contributes to this axis.

2. The idea of selecting part of the teacher's feature for debiased distillation under the meta learning framework is technically sound and intuitive.

3. Empirically, the method is effective ("Results show that BIRD improves the fairness of the knowledge distillation framework by 40.91%") and it is ready to be used along with existing KD methods to enhance the fairness of the distilled student.

**Weaknesses:**

1. One major problem with the proposed method is that it needs the internal features of the teacher model for distillation (Eq. 4). However, as the paper mentioned in the motivation, most of the FMs in the real world only provide APIs. I.e., their features are barely accessible. This limitation seems to undermine the practical value of the proposed method severely.

2. Another concern is that the fairness is improved but in many cases at a price of degraded accuracy. E.g., CLIP- ResNet50 with UTKFace in Tab. 1， CLIP-ViT-32 −→ResNet18 with UTKFace in Tab. 2, CLIP-R50 −→ResNet18 with UTKFace in Tab. 2. Namely, the proposed method is not very strong. The fairness issue may be a concern, while accuracy also matters.

A side concern is that, as seen above, the method does not perform well on the UTKFace dataset. Why?

3. Presentation: Some of the results are mistakenly highlighted. In Tab. 1, ∆mean-DEO, the highlighted results are sometimes not the best, which are quite confusing.

4. Minor issues.
- This paper seems quite relevant: https://aclanthology.org/2022.gebnlp-1.27.pdf. It reports a similar observation to Sec. 5.2 that KD amplifies biases.

**Questions:**

NAN

---

> ### Author Response · Authors · 2023-11-17
> **Rebuttal response to Reviewer qaWv (Part 1/2)**
>
> Thank you for your helpful feedback and for acknowledging our framework as intuitive and sound. We appreciate that you found our experiments extensive and method effective. Below, we address all your questions/concerns.
>
> **Practical Value of BIRD (Need for internal features)**
>
> We would like to clarify a potential misunderstanding here. Most cloud service provider (e.g., OpenAI, Google, and Amazon) pre-train a general-purpose feature extractor (called an encoder) and deploys it as a cloud service called an **Encoder As a Service (EAS)** [1,2,3,4], where a client/end-users queries the cloud service APIs for the final layer feature embeddings of its training/testing inputs when training/testing a downstream classifier. For safety and security-critical applications, a client aims to build an accurate and fair downstream classifier to safeguard algorithmic decisions and promote fairness. Further, Knowledge Distillation of Foundational models is usually performed at two ends: 1) developers who train student models and have access to everything about the model and 2) end-users who use the foundation model as a service (EAS).
>
> Further, we would like to clarify that **we do not use the internal feature representations of the foundation models** in Eqn. 4. Following the premise of EAS, BIRD **only utilizes the final layer representation of the models** and trains a single one-dimensional feature vector to re-weigh the teacher representations using FairDistill operator and improve the fairness of the distilled student model. Hence, using our proposed framework, an end-user can leverage the foundational model representations (from APIs) and use BIRD to distill its knowledge in a student network they are building to develop an accurate and fairer model.
>
> **Importance of Fairness and its Tradeoff with Predictive Performance**
>
> Studying fairness and its trade-off with predictive performance is imperative within the context of AI regulations and AI executive orders passed throughout the world. While accuracy is crucial, addressing fairness concerns is equally vital to prevent discriminatory outcomes. AI regulations like **the General Data Protection Regulation (GDPR) in the EU, the Personal Information Protection and Electronic Documents Act (PIPEDA) in Canada, the AI Executive Order in the USA, and the Data Protection Act (DPA) in the UK**, increasingly emphasize the importance of fairness, demanding that AI researchers and developers strike a balance between accurate predictions and equitable treatment of individuals. Executive orders underscore the need for transparency and accountability in AI systems, urging a thorough examination of how fairness considerations may impact predictive performance. Focusing solely on accuracy without regard for fairness can lead to biased outcomes, legal repercussions, and erosion of public trust in AI technologies, emphasizing the urgency of a nuanced understanding and integration of fairness principles into AI development practices.
>
> **Performance on UTKFace Dataset**
>
> We thank the reviewer for bringing up this insightful observation. We would like to report that BIRD, on average, **improves** fairness by **58.78%** (DEO-max) and **7.31%** (DEO-mean), while the F1 score **decreases** by **1.27%** and AUROC decreases by **6.34%** on the UTKFace dataset (see Table 1-2) compared to the baseline. Our results show that BIRD improves model fairness without sacrificing its predictive performance.
>
> Further, UTKFace is a small dataset that contains only 20,000 facial images with four ethnicities as opposed to the 200,000 images in the CelebA dataset. We hypothesize that one possible reason for the relatively poor performance of BIRD  on the UTKFace dataset could be the limited training dataset size. To this end, in response to the reviewers’ feedback, we perform an additional experiment and run BIRD on the UTKFace dataset with image augmentations (randomly flip the image horizontally, apply color jitter (using a $\sigma=0.2$), randomly rotate the image by 15 degrees and apply the random affine transformation). On average, we observe an increase of **+0.36% in AUROC** and **+1.17% in F1-score** for approximately the same fairness performance. Another notable observation is the massive performance boost when UTK augmentation is used with Clip-RN50 teacher – **+0.51% in AUROC** and **+1.53% in F1-score**, **-6.40% in DEO-mean**, and **-11.86% in DEO-max**. We added the exact ablation numbers in the appendix of the updated manuscript (see Table 10).

---

> ### Author Response · Authors · 2023-11-17
> **Rebuttal response to Reviewer qaWv (Part 2/2)**
>
> **Mistakenly highlighted results in Table 1**
>
> Thank you for pointing this out. We apologize for the confusion and have made the appropriate changes in the updated version of the draft.
>
> **Relevant related work**
>
> Thank you for sharing this work. We have cited and discussed the work in the revised related work section in our updated draft.
>
> We are very grateful to the reviewer for all their suggestions, as they have helped us improve our paper significantly. We tried to incorporate all the reviewer suggestions in our write-up and include all clarifications in the revised version. We would kindly request the reviewer to consider increasing their score.
>
> **References**
>
> [1] Qu, W., Jia, J., & Gong, N. Z. REaaS: Enabling Adversarially Robust Downstream Classifiers via Robust Encoder as a Service. arXiv, 2023.
>
> [2] https://clarifai.com/clarifai/main/models/travel-embedding
>
> [3] https://clarifai.com/clarifai/main/models/face-identification-transfer-learn
>
> [4] https://learn.microsoft.com/en-us/azure/ai-services/computer-vision/how-to/image-retrieval

---

> > ### Author Response · Authors · 2023-11-20
> > **Looking forward to your response**
> >
> > Dear reviewer,
> >
> > Thank you again for your thoughtful feedback. Following your suggestions, we provided a detailed clarification about the practical implications of BIRD, the importance of studying fairness and its trade-off with accuracy, and the results on the UTKFace dataset. We would love to hear your thoughts on our response. Please let us know if there is anything else we can do to address your comments.

---

> ### Comment · Reviewer_qaWv · 2023-11-22
> **Thanks for the feedback!**
>
> 1. The concern regarding the need for internal features is well resolved. Thanks for the clarification!
>
> 2. Meanwhile, about the accuracy vs. fairness tradeoff. I agree that fairness is very important, but it cannot explain why it must cost predictive performance in this work.
>
> 2.1 After reading the comments of other reviewers (e.g., QvRs), I also have the concern about using the traditional metric for evaluating predictive performance. If we look at Tab. 12, BIRD underperforms many other methods in terms of Top1 accuracy (and F-1 score), such as BKD and FitNet. I do not understand why the authors still claim their method "retains the predictive power (Top1-Acc, F1-score)".
>
> 2.2 Tab. 10. As seen, using Aug generally improves the predictive performance indeed, while in quite many cases (more than half of the presented results) Aug worsens the fairness metrics. So again, we see a tradeoff between predictive performance and fairness. I do not think the results can justify the claim in the caption "*BIRD while keeping the fairness criterion same as w/o Augmentation.*"
>
> Another problem with this table is, the results of the other methods are missing. We need comparisons to see if the method is really effective *against the counterparts*.
>
> ----
> A suggestion in rebuttal: If you revise the paper, it is highly suggested to **use some color to highlight the revised part**.

---

> ### Author Response · Authors · 2023-11-22
> **Response to Rebuttal**
>
> We are glad to hear that we were able to clarify your concerns regarding the use of internal features. Below, we address your remaining concerns.
>
> **Concern about using the traditional metric for evaluating predictive performance**
>
> We are excited to hear that the reviewer looked into our responses for other reviewers. In the light of using traditional metrics, we would like to clarify a potential misunderstanding. Classical computer vision benchmark datasets for image classification like CIFAR-10, CIFAR-100, and ImageNet used evaluation metrics like top-k accuracy (with k={1, 5} used in most papers) because these are balanced datasets, where each class had an equal number of images. However, fairness benchmark datasets like UTKFace and CelebA used in fairness work in computer vision are not well-curated and imbalanced datasets. Hence, it is common to use more robust evaluation metrics, such as AUROC and F1-score, when comparing predictive performance on these datasets.
>
> **Fairness vs Accuracy tradeoff in BIRD in Table 12**
>
> Thank you for the great point! In response to the reviewers' feedback, we analyzed the results in Table 12 and found that on average across two datasets, two teacher-student pairs, and three knowledge distillation baselines (MFD, BKD, and FitNet), BIRD achieves a significant improvement in fairness (**41.71% in DEO-mean and 32.13% in DEO-max**) and only takes a small hit in the predictive performance (**-1.67% in Top-1 accuracy and -2.37% in F1-score**). Thank you for highlighting the misleading statement and we are happy to revise our statement. We intended to say that BIRD demonstrates that its predictive performance is closer to baseline methods and deem this as a new frontier in exploring the fairness-accuracy tradeoff, where we take a small hit in the accuracy and achieve $\approx$**25**$\times$ higher fairness performance.
>
> |                             | **BIRD** | **MFD** | **FitNet** | **BKD** |  **AD** |
> |-----------------------------|:--------:|:-------:|:----------:|:-------:|:-------:|
> | **Top-1 ($\uparrow$)**      |   77.68  |  77.28 |   79.88   | 79.88 | 68.45 |
> | **F1-Score ($\uparrow$)**   |   77.44  | 77.22 |   80.86  |  79.98 | 67.94 |
> | **DEO-mean ($\downarrow$)** |  11.63 | 19.32 |   19.92   |  20.66 | 21.32 |
> | **DEO-max ($\downarrow$)**  |  20.97 |  30.86 |    30.19   |   31.7  |  35.25  |
>
> **Comparisons to see if the method is really effective against the counterparts in Table 10**
>
> We thank you for looking into these results in detail. We would like to clarify that we ran the augmentation experiment in response to your comment during the rebuttal phase and finished the BIRD experiments by the time of our initial rebuttal response. We have started running new experiments on computing the predictive and fairness performance of baseline methods and comparing them to the augmentation results of Table 10 and would like to request an additional 4-5 hours as we expect the baseline results to be complete by then.
>
> **A suggestion in rebuttal: If you revise the paper, it is highly suggested to use some color to highlight the revised part**
>
> Thank you for your suggestion. We marked the changes in blue in the main tex of the revised manuscript and, in response to the reviewer's suggestion, we have marked the caption of the revised tables of the appendix in blue too in the updated draft of the manuscript.
>
> Please let us know if you have any further questions or concerns.

---

> > ### Comment · Reviewer_qaWv · 2023-11-22
> > **Further discussion**
> >
> > (1) The authors state:
> >
> > * *“BIRD achieves a significant improvement in fairness (+41.71% in DEO-mean and +32.13% in DEO-max) and only takes a small hit in the predictive performance (-1.67% in Top-1 accuracy and -2.37% in F1-score). "*
> > * *"where we take a small hit in the accuracy and achieve ~25x higher fairness performance."*
> >
> > Clarification needed - How do you get these numbers: "41.71%, 32.13%, -1.67%, -2.37%, 25x". These are compared against which method?
> >
> > (2) The presented new table in the response above - Are they new results? I do not find it in the paper and supplementary. What are they supposed to show?

---

> > > ### Author Response · Authors · 2023-11-22
> > > **Rebuttal Response**
> > >
> > > We thank the reviewer for the prompt response and apologize for the confusion in our previous response. We would like to point to the reviewer that the mentioned table is a summary of Table 12, which was pointed out by the reviewer in their earlier response. Please see below for the details of how we generated the numbers in the following table.
> > >
> > > |                             | **BIRD** | **MFD** | **FitNet** | **BKD** |  **AD** |
> > > |-----------------------------|:--------:|:-------:|:----------:|:-------:|:-------:|
> > > | **Top-1 ($\uparrow$)**      |   77.68  |  77.28 |   79.88   | 79.88 | 68.45 |
> > > | **F1-Score ($\uparrow$)**   |   77.44  | 77.22 |   80.86  |  79.98 | 67.94 |
> > > | **DEO-mean ($\downarrow$)** |  11.63 | 19.32 |   19.92   |  20.66 | 21.32 |
> > > | **DEO-max ($\downarrow$)**  |  20.97 |  30.86 |    30.19   |   31.7  |  35.25  |
> > >
> > > Each cell of this table is averaged over the two models (CLIP-ViT-32 →ResNet18 and CLIP-R50 →ResNet18) and two datasets (CelebA and UTKFace), i.e., a total of four settings to analyze the overall performance of BIRD and baselines. For instance, BIRD Top-1 is the average of Top-1 scores over four reported numbers of BIRD in Table 12.
> > >
> > > **Clarification needed - How do you get these numbers: "41.71%, 32.13%, -1.67%, -2.37%, 25x". These are compared against which method?**
> > >
> > > We use the above table to compute the relative improvement in the performance of BIRD over other baseline methods. More specifically, we obtain the relative percentage change of BIRD with each method $m$ as:
> > >
> > > (Performance of BIRD in the above table - Performance of $m$ in the above table) / Performance of $m$ in the above table
> > >
> > > and get:
> > >
> > > |                             | **w/ MFD** | **w/ FitNet** | **w/ BKD** | **Average** |
> > > |-----------------------------|------------|---------------|------------|-------------|
> > > | **Top-1 ($\uparrow$)**      |      0.51% |        -2.76% |     -2.76% |      -1.67% |
> > > | **F1-Score ($\uparrow$)**   |      0.28% |        -4.23% |     -3.17% |      -2.37% |
> > > | **DEO-mean ($\downarrow$)** |     39.81% |        41.61% |     43.71% |      41.71% |
> > > | **DEO-max ($\downarrow$)**  |     32.03% |        30.53% |     33.84% |      32.13% |
> > >
> > > We now take the average of the change with respect to MFD, FiTNet, and BKD (the competitive baselines) to report the numbers: 41.71%, 32.13%, -1.67%, -2.37%. For the 25x number, it's **41.71/1.67 = 24.98 $\approx 25$**
> > >
> > > We hope these clarifications help you in understanding our results better. We will add these details in the next revised version of the draft. Please let us know if you have more questions.

---

> > > > ### Author Response · Authors · 2023-11-23
> > > > **Augmentation results for baselines**
> > > >
> > > > We are excited to share that our augmentation results using baselines have just been completed. In response to the reviewer's feedback, we added **Table 14** to the revised manuscript and shared the results of baselines and BIRD using image augmentation during the training. Our preliminary results show that, on average across four teacher-student pairs (CLIP/ViT-32 → CLIP/ViT-32, CLIP/RN50 → CLIP/RN50, CLIP/ViT-32 → ResNet18, and CLIP/RN50 → ResNet18), BIRD achieves an improvement in fairness (**5.84% in DEO-mean and 14% in DEO-max**) and only takes a minor hit in the predictive performance (**-1.22% in AUROC and -1.28% in F1-score**). Note that these numbers are computed using the strategy detailed in our earlier response.
> > > >
> > > > We apologize for the delay in sharing the above results and are happy to answer any further questions.

---

> > > > ### Comment · Reviewer_qaWv · 2023-12-03
> > > > **last comment**
> > > >
> > > > Thanks for the clarification, but this metric definition (e.g., how the 25x is calculated) is sort of new - I did not see similar definitions in other papers. When the authors say "their method achieves 25x higher fairness performance", it is hard to know if this 25 is significant or not, simply because it is not a well-accepted metric (proposed by this work). I am quite reserved about whether we should average the results of different datasets / networks to report the final results (the choices of datasets and networks will affect the results, right? - Then, what is the variation if I choose another set of datasets / networks?)
> > > >
> > > > What I am confident is that, the method poses an obvious tradeoff between accuracy and fairness. There is no clear evidence that the proposed method is much better than its counterparts. Rating thus maintained at 5.

---

### Official Review · Reviewer_Gu1U · 2023-12-03

**Soundness:** 3 good
**Presentation:** 2 fair
**Contribution:** 3 good
**Rating:** 6
**Confidence:** 4

**Summary:**

The authors adapt MAML  https://arxiv.org/abs/1703.03400 for improving fairness in knowledge distillation.

Overall, the authors use MAML to find a transformation on the teacher logits in a way which maximizes fairness *after* the student's update. This is analogous to the original use for MAML, which is to find model parameters such that the loss of the model *after* making an update on the target task is minimized.

To summarize the authors' algorithm, one iteration proceeds in the following steps: 1) find a gradient update on the student with fixed transformation on the teacher, 2) find transformation on the teacher optimizing for the fairness objective under the updated student, 3) forget the the updated student from step 1) but use the updated transformation from step 2), and find a new update for the student parameters when optimizing jointly for knowledge distillation, one-hot cross entropy and fairness objectives.

The authors provide experiments and ablations across several image benchmarks and report good results on improved fairness.

**Strengths:**

The problem of fairness that the paper considers is important. The algorithm is very interesting and thought provoking. The results look good. I appreciate that the authors conduct a few interesting ablations shedding some light on the algorithm.

**Weaknesses:**

The notation seems superfluous and the explanation for the algorithm is not very clear. I suggest the authors stick to the MAML notation as much as possible. For example, why do authors introduce f_copy in Figure 1? The original paper serves as an example that this is not necessary, and we can express everything mathematically using the original student parameter and the gradient updates. The FAIRDISTILL operator is also not very helpful, i think the Phi vector alone as a notation is sufficient.

After giving it some thought, the algorithm makes some intuitive sense to me. However, it wasn't very clear to me in the first reading. I think the authors should give some explanation and intuition why adapting MAML to distillation in the specific way as introduced by the authors should help with fairness.

Minor:

The authors refer to image models as foundation models, which is confusing, because the term means generative large language models.

'We leverage meta-learning frameworks consisting of two optimization steps' -> We leverage a meta-learning framework consisting of two optimization steps

**Questions:**

The observation that distillation harms fairness is not new, see https://arxiv.org/pdf/2106.10494.pdf section 3.7. It would be worthwhile reflecting this in the paper accordingly.

Can authors comment on a possible baseline directly optimizing for DEO (as introduced in the paper). A paper considering a related objective for distillation: https://proceedings.mlr.press/v216/wang23e/wang23e.pdf

'where L_reg is the regularization on f_S that penalizes student bias' -- is L_reg same as L_outer?

More baselines should be considered, e.g. what if we add equation 7 or 8 to the model objective? (is that the baseline from 5.2.4?) What if equation 6 is directly optimized for?

Why is it necessary to split the train dataset into a meta dataset and a new train dataset? What if one used the same (train) dataset for all the steps of the algorithm?

---

### Author Response · Authors · 2023-11-21
**Looking forward to your feedback**

Dear reviewers,

Thank you again for your precious time and valuable comments! Following your suggestions, we included new results, clarifications on the importance of fairness, the intuition behind choosing meta-learning, and the technical novelty of our learning-to-learn framework, in our rebuttal response. We would love to hear your thoughts on our responses. Please let us know if there is anything else we can do to address your comments.

Thank you very much!

Best regards,

Authors of Submission4352

---

### Meta-Review · Area_Chair_orYA · 2023-12-06

**Metareview:**

There was some disagreement in recommendation, albeit with no strong opinion either way. The main strengths and weaknesses identified were:
+ generic framework to remove bias in KD
+ technical novelty in the proposed solution
- results are on smaller-scale datasets
- introduction of a fairness-quality tradeoff

For the last two points, the response noted that the datasets used are standard in the fairness literature. They also argued that the degradation in quality is minor compared to the gains in fairness.

The AC sought an additional review which was leaning positive, albeit noting some critiques of the presentation and notation.

The AC also inspected the manuscript. We tend to agree that some parts of the paper could be much clearer. Some of the below could be inferred from context, but it is really incumbent upon the authors to make their paper as easily readable as possible. We would strongly advise the authors carefully incorporate the reviewers' many detailed suggestions to improve the clarity of the manuscript. Some specific comments:

- Sections 1-4 do not appear to explicitly state the notion of fairness under consideration (demographic parity, disparate impact, equality of opportunity, predictive parity, worst-subgroup fairness, counterfactual fairness, ...).

- the second term of Equation 1 refers to $\mathbf{\hat{y}}$ which is not precisely defined; is it the softmax probabilities for a *single* example? For the latter, it is unclear why there is no subscript $i$. It is also unclear why the first term is written as a summation over $i$, while the second term is written in apparent vectorized format. We similarly note that Equation 2 is missing an explicit summation over $i$.

- consider using a superscript for $y^p$, since the subscript notation $y_i$ is previously used to denote an individual sample. Alternately, use a different variable name. In fact, it seems the paper starts referring to $p$ as the protected attributes in Section 4.1. It is not clear at the outset that one has *multiple* protected attributes.

- the paper does not motivate *why* they follow a meta-learning approach. The first wave of approaches for both KD and fairness employ the regular statistical learning paradigm. It would be good to explain why one cannot adapt these standard ideas to incorporate fairness into the KD process.

- further to the above, there is limited discussion, conceptual or otherwise, of the weaknesses of existing approaches to tackle fairness in KD. It is mentioned on page 1 that a method of Jung et al. '21 does not demonstrate gains on large models, but that does not specify whether there are fundamental limits in the technique which necessitate a wholly new approach.

- in the FairDistill update section, $\theta_{\rm copy}$ does not appear to be explicitly defined (though it can be inferred from context).

- in Equation 5, it is unclear why the summation over $i$ goes to $C$ and not $N$.

- in Equation 5, it is unclear what the precise meaning is of $y_i \mid y_p = j$. In the prequel, it does not appear that $y_i$ is a random variable. Even interpreting it as a realisation of a draw from an underlying distribution, it is not clear how to interpret this quantity. The text following the equation talks about a "task attribute", but that term does not appear to be defined previously.

- in Equation 5, it is unclear what the arguments to the $\max( \cdot )$ operator are. It appears to have a *single* argument, namely the summation over $j$. This is not at all clear. Do the authors mean to have a maximum over $j$? Do they mean to have a maximum over $p$?

- in Equation 5, there ought to be a comment on how one gets stochastic gradients in the presence of a maximum over potentially multiple elements.

- in the response, the authors claim a 25x improvement in the fairness performance. This appears to be the ratio of the average % improvement in DEO-max and average % degradation in top-1 accuracy. It is advised to provide a citation of prior work which reports the ratio of DEO and accuracy to summarize fairness.

Given the above, and the reviewers' own critiques, the paper is a borderline case. In our opinion, the work could substantially benefit from changes to improve clarity which would better contextualize the work, and make it useful for future research to build upon. For this reason, our recommendation is that the authors incorporate these changes and undergo review in a future conference.

**Justification For Why Not Higher Score:**

Clarity of the paper can be significantly improved, including on important technical details and motivation of the particular approach.

**Justification For Why Not Lower Score:**

Some technical innovation, some promising results.

---

### Decision · Program_Chairs · 2024-01-16

Reject